# Interglacials of the Quaternary defined by northern hemispheric land ice distribution outside of Greenland

Peter Köhler 1✉ & Roderik S. W. van de Wal 2

Glacial/interglacial dynamics during the Quaternary were suggested to be mainly driven by obliquity (41-kyr periodicity), including irregularities during the last 1 Myr that resulted in on average 100-kyr cycles. Here, we investigate this so-called Mid-Pleistocene Transition via model-based deconvolution of benthic $\delta^{18}$O, redefining interglacials by lack of substantial northern hemispheric land ice outside of Greenland. We find that in 67%, 88% and 52% of the obliquity cycles during the early, middle and late Quaternary, respectively, a glacial termination is realized leading to irregular appearances of new interglacials during various parts of the last 2.6 Myr. This finding suggests that the proposed idea of terminations leading to new interglacials in the Quaternary as obliquity driven with growing influence of land ice volume on the timing of deglaciations during the last 1 Myr might be too simple. Alternatively, the land ice-based definition of interglacials needs revision if applied to the entire Quaternary.

[1] Alfred-Wegener-Institut Helmholtz-Zentrum für Polar-und Meeresforschung (AWI), P.O. Box 12 01 61, Bremerhaven 27515, Germany. [2] Institute for Marine and Atmospheric Research Utrecht (IMAU) and Faculty of Geosciences, Department of Physical Geography, Utrecht University, Princetonplein 5, Utrecht 3584 CC, The Netherlands. ✉email: peter.koehler@awi.de

Changes in Earth's orbit are understood to be the funda-
mental reasons for the ice age dynamics during the Qua-
ternary[1–3]. However, orbital theory alone cannot explain
the reconstructed evolution from a 41-kyr dominated glacial
cyclicity earlier in the Quaternary towards a 100-kyr periodicity
during the last 1 Myr[4,5]. For an explanation of this so-called Mid-
Pleistocene Transition (MPT), feedbacks in the climate system
have to be considered. Prominent candidates of those feedbacks
are land ice dynamics[6,7], possible driven by regolith removal[8,9]
and the gradual decline in atmospheric $CO_2$ concentrations
during glacial minima[10–12]. Hypotheses to explain this MPT
have been the subject of many studies throughout the last four
decades[6,8,9,13–26]. One of the most recent approaches to explain
the observed pattern during the MPT[24] (T17 in the following)
developed a simple rule by which the onset of new interglacial
periods during the last 2.6 Myr are identified based on caloric
summer half-year insolation at 65°N, whereby it assumes that
deglaciations (melting of land ice) mainly depend on local sum-
mer insolation. In this concept, the amount of energy necessary to
trigger a deglaciation which leads into an interglacial decreases
with the amount of time since the previous deglaciation as it is
assumed in T17 that ice-sheet instability increases with time.

In T17, interglacials across the whole Quaternary are defined
by passing thresholds in the detrended benthic LR04 $\delta^{18}O$ stack[4],
including some tests of these methods with alternative benthic
$\delta^{18}O$ records. This definition of interglacials agrees for the last
800 kyr with findings of a more rigorous community effort[27]
based on the analysis of various climate variables. As consequence
of this new definition of interglacials[24], no regular 100-kyr peri-
odicity of interglacials occurs after the MPT. Instead, during th-
e last ~1 Myr, the regular appearance of interglacials every 41-kyr
following the pacing of obliquity before the MPT is replaced
by an irregular appearance of interglacials with ~80-kyr or
~120-kyr cyclicity of interglacials (Fig. 1). This finding of irre-
gular interglacials is very similar to a concept[20] of an increasing
number of skipped obliquity-driven terminations over time.

Other studies tried to explain the MPT from a more enhanced
glaciological perspective[6,8,9,28]. One study[6] deconvolved the same
benthic $\delta^{18}O$ stack[4] used in T17 into its sea level and temperature
components by applying physically consistent 3-D land ice
models and came up with the simulated temporal evolution of
land ice distribution. They concluded that after the MPT the
longer glacial cycles with larger amplitudes in sea-level change
seemed to be controlled by North American land ice dynamics:
when growing to a certain size, the Laurentide and Corrdilleran
ice sheets merge to a single dome, which supports thicker ice

sheets. At the same time the thicker ice leads to a decoupling of
the temperature at the ice surface and ice base. This results in a
larger fraction of the ice sheet to be at pressure-melting point and
allows sliding of the ice over the bed and facilitating the degla-
ciation of the North American ice-sheet complex.

While those hypotheses mainly based on orbital theory[19,20,24]
were very successful in defining a rule by which deglaciations or
interglacials might have been triggered, they had only little
emphasis on the details of the glaciations, even though the land
ice albedo feedback on climate is probably essential to understand
the MPT. Changes in ice volume or global mean sea level—
necessary to define interglacial periods—were approximated from
deep ocean $\delta^{18}O$ or simple equations[20,24], but seldom considered
in more detail. This reduction in land ice complexity is also
underlying the restriction of most orbital theories to an in-depth
analysis of a single latitude, mainly 65°N.

In the following, we will analyse land ice dynamics obtained
in the framework of the inverse deconvolution of the LR04 $\delta^{18}O$
stack[6,29]. Although our ice dynamics are the outcome of a 3-D
land ice model, we understand them to be more a model-based
interpretation of proxy data rather than a fully independent
simulation. We apply the absence of substantial northern
hemispheric land ice outside Greenland as a criterion for the
definition of interglacials across the Quaternary. When applied
for the last 800 kyr this definition has been shown[27] to be
robust against a range of definitions and thresholds. Our ana-
lysis avoids a discussion of benefits and shortcomings of the
various insolation metrics (e.g., the local northern summer
insolation intensity[14], the integrated summer insolation[19], or
caloric summer half-year insolation[24]). The latitudinal dis-
tribution of land ice across the northern hemisphere would
need to include insolation at all latitudes and, if investigated
within an energy balance scheme[30], across the full year. Our
findings will be discussed in the light of uncertainties and
available independent evidences, including the role of atmo-
spheric $CO_2$ concentration. We find that the glacial/interglacial
amplitudes of northern hemisphere land ice variability outside
of Greenland are much smaller prior to than after the MPT. At
the same time the contributions from deep ocean temperature
and ice volume in Antarctica and Greenland to this change in
glacial/interglacial amplitude across the MPT in the corre-
sponding benthic $\delta^{18}O$ stack are much smaller. According to
our new definition of interglacials their appearance is in general
following the 41-kyr cycle of obliquity with various exceptions
throughout the last 2.6 Myr, suggesting a more complex phy-
sical mechanism triggering the glacial terminations.

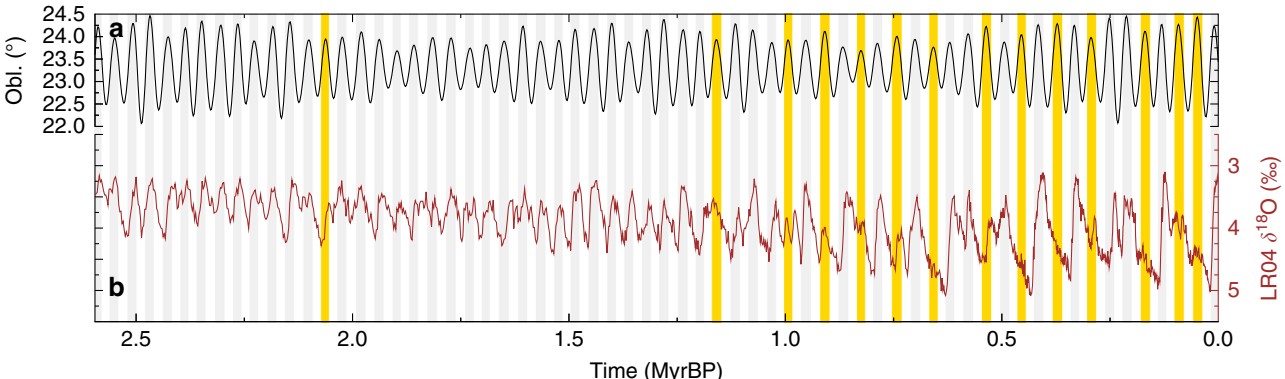

**Fig. 1 Obliquity and climate dynamics during the Quaternary. a** Obliquity[64]. Obliquity cycles with (without) associated new interglacials as defined in T17
are labelled with grey (gold) vertical bands for times when obliquity is above average (>23.3°). **b** Integrated climate change contained in the LR04[4] benthic
$\delta^{18}O$ stack (brown line) plotted on an inverse y-axis.

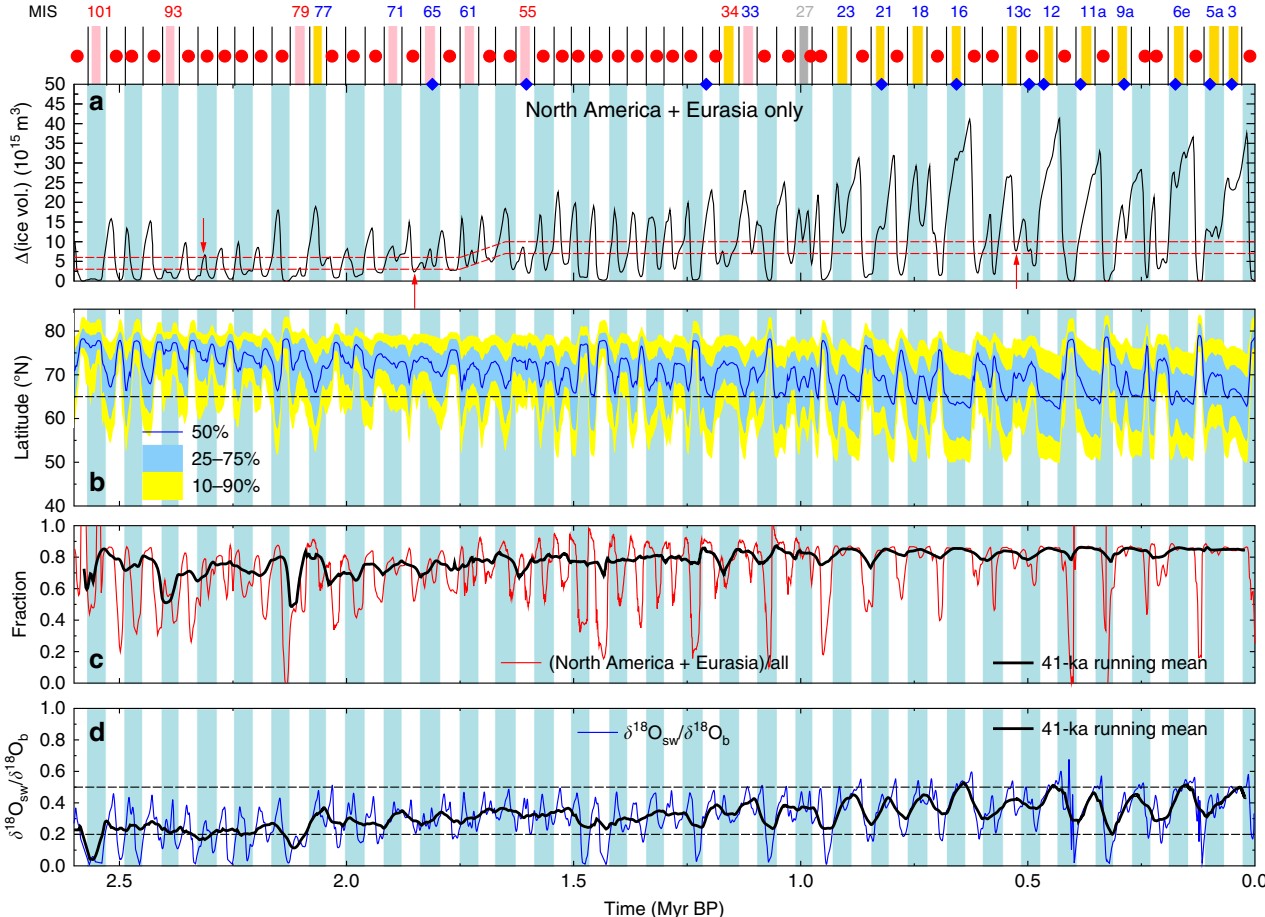

**Fig. 2 Defining interglacials by land ice volume.** For easy identification of different obliquity cycles, every second cycle (minima-to-minima) is colour coded in light blue bars. **a** Ice-volume change in North America and Eurasia used to define interglacials. Two thresholds (broken lines) need to get crossed by the ice volume change to identify an individual interglacial. Vertical arrows mark where and how the thresholds have been constrained by our ice-volume results. Obliquity minima are marked by vertical lines above (**a**). The onset of a new interglacial is marked (red circles), obliquity cycles without new interglacials are colour coded with vertical bars (gold: already found in T17; pink: new in this study: dark grey: found in T17, but not confirmed in this study). Marine isotope stages (MIS) in obliquity cycles without a new interglacial are labelled for skipped terminations (blue), or continued interglacials (red). Blue diamonds show obliquity cycles with skipped terminations (and therefore no interglacials) during the last 2 Myr after a different study[20]. **b** Ice volume in North America and Eurasia as function of latitude. Dark blue line shows the position of the 50% line, implying that half of the ice volume is situated either north or south of it. Yellow (light blue) area shows the latitudinal position of 10–90% (25–75%) of the ice volume. **c** The fraction of change in North America and Eurasia ice volume in relation to ice-volume changes globally. **d** Fraction of $\delta^{18}O$ caused by ice-volume (sea level) changes. Seawater $\delta^{18}O_{SW}$ is the ice-volume contribution to benthic $\delta^{18}O_b$ as deconvolved earlier[29]. To guide the eye, two broken lines at 0.2 and 0.5 are included.

## Results

**Defining interglacials by land ice distribution.** Despite the notation that interglacials are typically characterized by little land ice left in the northern hemisphere outside Greenland, T17 characterized interglacials (beyond the last 800 kyr that have been covered in an in-depth analysis of various climate variables[27]) by passing thresholds in the benthic LR04 $\delta^{18}O$ stack, which has been detrended before 1.5 Myr BP. Because benthic $\delta^{18}O$ is globally integrating all land ice changes, also on Greenland and in Antarctica, and because it additionally contains a contribution from deep ocean temperature change, the record has been detrended in T17 in order to approximately correct for these secondary effects. Here, we are in the position to define inter-glacials as originally meant in T17 and as proposed as an objective criterion for the last 800 kyr[27]: by changes in land ice volume in North America and Eurasia (Fig. 2a). One needs to be aware that both approaches heavily rely on the LR04 $\delta^{18}O$ stack as a valid representation of the true mean deep ocean $\delta^{18}O$. Thus, any bias contained in it, e.g., due to the stacking method, spatial

heterogeneity, or undersampling will also be contained in both T17 and our interpretation here.

We use, as in T17, two thresholds that need to be crossed by the integrated land ice volume located in North America and Eurasia to define an individual interglacial (Fig. 2a): the lower for flagging a new interglacial, the upper for separating two interglacials by a glacial period in-between. The vertical arrows in Fig. 2a constrain our thresholds. They have been chosen to replicate the detected new interglacials during the last 800 kyr[27] and to maximize the number of obliquity cycles that contain the onset of an interglacial earlier. We put little emphasis on the relative timing of the new interglacials with respect to incoming insolation because in full year insolation that also varies as a function of latitude no maxima in insolation can be simply defined. Thus, we only place the detected onsets of interglacials into the corresponding obliquity cycles (markers above Fig. 2a). See "Methods" for details.

For times after 1.6 Myr BP, our approach detects nearly the same obliquity cycles as in T17 with the onsets of new

interglacials. This includes 13 obliquity cycles without detected interglacials, of which 12 are known as "skipped terminations," while one (including MIS 34) is a continued interglacial. The only two exceptions are periods containing MIS 27 (~1 Myr BP) and MIS 33 (~1.11 Myr BP). According to our definition, the obliquity cycle around 1 Myr ago should be classified as interglacial, but it fails (although very closely) to be so in T17. Around 1.11 Myr BP, including MIS 33, the ice volume very closely fails to meet the lower threshold to be defined as interglacial. A slightly higher threshold leading to an interglacial here, in agreement with T17, would then also classify the period around MIS 13c (550 kyr BP) as interglacial, which would then disagree with the previous classification[27].

In order to minimize the number of obliquity cycles without new interglacials, we had to lower the thresholds by which we detect interglacials for earlier time periods in the Quaternary. This change in the thresholds has a similar effect as the detrending of $\delta^{18}O$ prior to 1.5 Myr BP in T17 and can be interpreted as a change in the background climate by, e.g., a long-term change in $CO_2$ concentration. However, there are still seven periods that have been identified as interglacials in T17 and that do not show up as interglacials when our criterion is applied (pink bars above Fig. 2a). The obliquity cycle without a new interglacial around MIS 61 is the only one that is situated in the time window in which the thresholds are changing. A slightly revised definition of these changes in the thresholds, which would not influence any of the interglacial classifications elsewhere, might lead to different results here. If we choose alternative thresholds by which we define interglacials, we increase the amount of obliquity cycles without new interglacials. If we keep the thresholds constant throughout the Quaternary at either the lower or higher levels detected for the early or later Quaternary (Fig. 3b, c), 4–8 more obliquity cycles are found without new interglacials.

Thus, when using the occurance of little land ice in the northern hemisphere outside of Greenland as criteria for interglacials, we find that not only during the last 1 Myr but also from 2.6 to 1.6 Myr BP prior to the MPT the occurence of new interglacials is rather irregular. In four periods before the MPT (including the red-labelled MIS 55, 79, 93, 101, Fig. 4), the ice volume was smaller than necessary to flag for a glacial, thus marking those periods as extended, or continued, interglacials, while four obliquity cycles with skipped terminations exist (ice volume is too large to flag for a new interglacial, blue-labelled MIS 61, 65, 71, 77 in Fig. 4). The Mid-Pleistocene between ~1 and 1.6 Myr BP (MIS 25–54) is a time window with regular glacial/interglacial dynamics, consisting of 13 obliquity cycles with a new interglacial each. The only exception is a period between 1.2 and 1.1 Myr BP, in which the onset of two successive interglacials is more than 100 kyr apart, caused by a skipped termination (MIS 33) that follows on a continued interglacial (MIS 34), a pattern also found around 2.1 Myr BP. The calculated mean ($\pm 1\sigma$) interglacial return times would be 60 ($\pm 22$) kyr, 47 ($\pm 19$) kyr, and 79 ($\pm 24$) kyr for the pre-MPT, MPT, and post-MPT, respectively. However, similarly as the average 100-kyr periodicity in the post-MPT, that according to the idea of an obliquity-driven system with skipped termination causing either a ~80 or ~120 kyr return time of interglacials[20], these mean values here also suggest some misleading average periodicities. A more helpful way seems to be to count the realized new interglacials with respect to the potential new interglacials, with the later being identical to the obliquity cycles. Following this idea we find a new interglacial (or termination) realization fraction of 67% in the pre-MPT (16 out 24, MIS 55–101), of 88% in the MPT (14 out of 16, MIS 27–53), and of 52% in the post-MPT (12 out 23, MIS 1–25). These fractions should not be mixed with the fraction of obliquity cycles that include interglacials, since the

realization of new interglacials decreases not only for obliquity cycles with a skipped termination but also for those with a continued interglacial.

When our approach is based on the alternative thresholds (Fig. 3b, c) we also find that the reason for an obliquity cycle to miss a new interglacial might change. When based on the higher threshold (Fig. 3c) we only find continued interglacials (red-labelled MIS), but no skipped terminations (blue-labelled MIS) for times earlier than MIS 33 around 1.15 Myr BP. Furthermore, the result for MIS 34 largely depends on the chosen threshold. The changes in ice volume between a glacial and an interglacial period in the early Quaternary are rather small and definitions based upon will always be prone to some arbitrariness.

**Comparing with other definitions of interglacials.** How robust is our classification, especially when it differs from T17? In addition to the problematic case of MIS 61 discussed above, we find in MIS 65 an ice volume that is very close to the chosen threshold. However, MIS 65 (together with our new different finding in MIS 55) has already been detected as one of the obliquity cycles of the last 2 Myr with a skipped termination[20], although our analysis detects MIS 55 as a continued interglacial. MIS 71, which is in our classification clearly a skipped termination, has in T17 been classified as an uncertain interglacial; its caloric summer insolation was just below the deglaciation threshold, while its effective energy was just above it. Here, effective energy is peak caloric summer insolation plus a term related to the elapsed time since the last full deglaciation that accounts for the increasing ice-sheet instability. In summary, from the seven differences in the detection of interglacials in the Early Quaternary between T17 and this study, 2–3 might be questionable, leaving a residual of 4–5 robust differences between both approaches.

The large number of obliquity cycles without new interglacials prior to the MPT in our results, which did not show up in T17, can be understood as follows: The deconvolution[29] of benthic $\delta^{18}O$ showed that prior to the MPT only 20% of the long-term signal (41-kyr running mean) has been caused by sea level or ice volume ($\delta^{18}O_{sw}$), rising to 50% during glacial maxima after the MPT, while the dominant part of the change in benthic $\delta^{18}O$ has been caused by ocean temperature variations (Fig. 2d). Furthermore, the fraction of ice volume changes based on North America and Eurasia was falling to 50% during sustained long interglacials prior to the MPT, while it stayed around 85% during the last 1 Myr (Fig. 2c). In other words, prior to the MPT the relatively large anomalies in the LR04 benthic $\delta^{18}O$ (from which a regular pattern of interglacials has been detected in T17) transform into rather small anomalies in the northern hemispheric ice-volume changes outside of Greenland. Thus, the detrending of the LR04 $\delta^{18}O$ applied in T17 is according to our results too imprecise to correct as suggested for the effects of deep ocean temperature and ice volume in Antarctica and Greenland (Fig. 4b). By our definition, this turns the obliquity cycles containing MIS 55, 61, 65, 71, 79, 93, and 101 into periods without the onset of a new interglacial.

Skipped terminations identified previously for the last 2 Myr[20] are by definition related to an obliquity cycle without an interglacial. Therefore, they can be considered as a negative interglacial detector (blue diamonds above Fig. 2a), and we find a huge overlap between the different approaches, but not identical results. This alternative analysis[20] was based on benthic $\delta^{18}O$ records different from those used in T17, plotted on an age model which is independent of orbital assumptions to avoid circular reasoning and solely based on an own definition of terminations. Thus, identified terminations (that subsequently led into a new interglacial) or skipped terminations (that failed to lead to a new interglacial) are in detail based on different criteria.

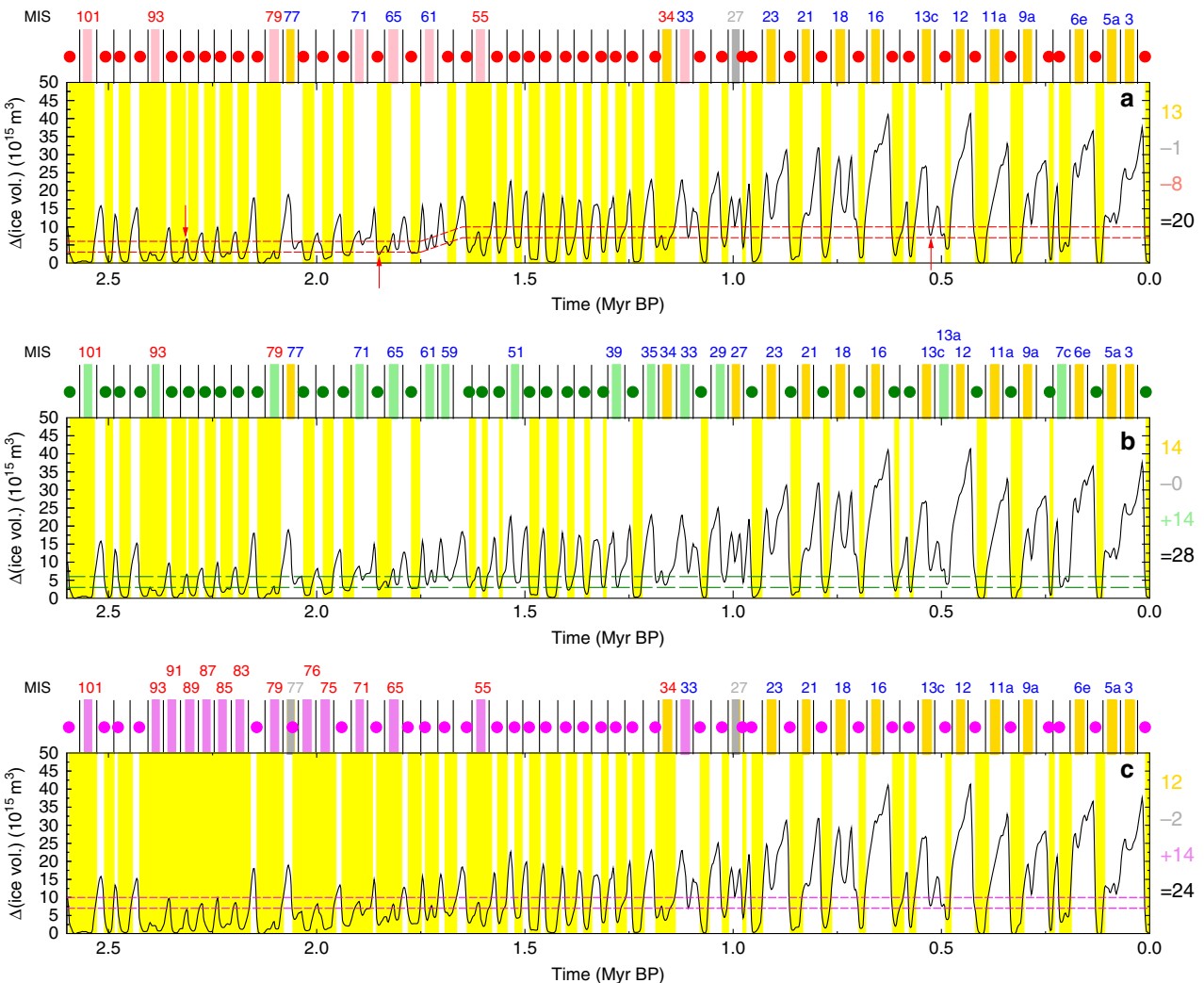

**Fig. 3 Alternative definitions of Quaternary interglacials based on land ice volume.** Ice volume changes in North America and Eurasia are used to define interglacials. Different sets of two thresholds (broken lines) need to get crossed by the ice-volume change to identify an individual interglacial, see text for details. On top of each panel, obliquity minima (vertical lines) and the onset of each new interglacial are marked by circles, when the plotted ice volume is crossing the lower of both thresholds after having been in glacial mode (above the higher of both thresholds). Marine isotope stages (MIS) in obliquity cycles without a new interglacial are labelled for skipped terminations (blue), or continued interglacials (red). Yellow background colours mark the times from the beginning of a new interglacial to the beginning of a new glacial. Colour bars above the panels mark obliquity cycles which are without a new interglacial (gold: according to T17; pink/lightgreen/violet: additional interglacials according to this study; grey: contains interglacials according to T17, but not confirmed in our definition). Numbers on the right summarize obliquity cycles without interglacials according to this colour code. **a** Standard definition of thresholds, minimizing the number of obliquity cycles without interglacials (similar to Fig. 2a). Time-independent definition of thresholds using values constrained (**b**) for the early or for (**c**) the late part of the Quaternary.

In combination with our own effort to identify interglacials based on the lack of northern hemisphere land ice outside of Greenland, the different definitions of interglacials converge to a unique solution across most of the last 1.6 Myr, but not across the whole of the Quaternary. Between 1.6 and 2.6 Myr BP, the eight (out of 24) obliquity cycles without the onset of a new interglacial suggests that our classification—based on a deconvolution of the LR04 $\delta^{18}$O stack with currently available information—does not follow a simple rule. Instead, our findings suggest that across various parts of the Quaternary terminations leading to new interglacials are more irregularly placed and not necessarily forced by obliquity changes as is often assumed. This conclusion is conditional to the definition of an interglacial as having little ice in the Northern Hemisphere outside Greenland. Thus, alternative definitions of interglacials (as reviewed previously[27]) might be necessary when applied to the early Quaternary, and potentially

also to earlier periods during the Neogene. Actually, our definition of interglacials clearly illustrates a shift in the climate system, from an interglacial-dominated period in the early Quaternary to a glacial-dominated period in the late Quaternary, a pattern which has originally been suggested from results of a simplified vertically integrated sectorial ice model without thermodynamics[28]. This becomes especially clear when we base our findings on constant thresholds (e.g., Fig. 3c).

## Discussions

Most recently, an intermediate complex Earth system model[9] was forced solely by known changes in the orbital parameters and some assumptions on long-term changes in atmospheric $CO_2$. The authors were able to simulate climate dynamics across the MPT confirming a dominant role of regolith removal on glacial

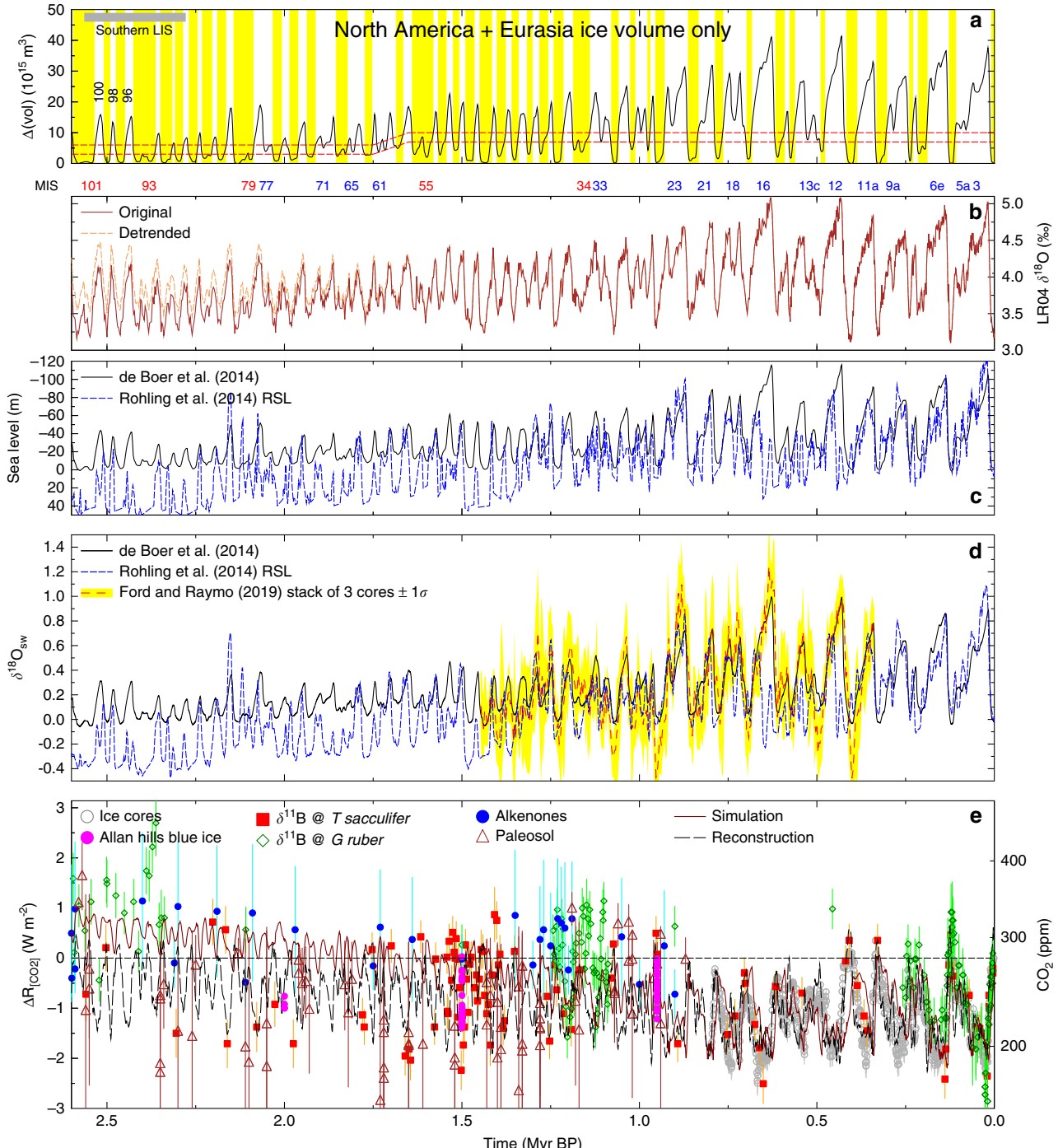

**Fig. 4 Knowns and unknowns of Quaternary climate. a** Ice volume change in North America and Eurasia used to define interglacials (see caption to Fig. 2a for details). Earliest southern advance of the Laurentide ice sheet to 39°N in Missouri marked by grey horizontal bar[46]. Prominent first large glaciations in the Quaternary labelled with their MIS 100, 98, 96. Marine isotope stages (MIS) in obliquity cycles without a new interglacial are labelled for skipped terminations (blue), or continued interglacials (red). **b** LR04 benthic $\delta^{18}O$ stack[4], original and as detrended in T17. **c** Sea level and **d** $\delta^{18}O_{sw}$ of different studies[29,48,49]. **e** Atmospheric $CO_2$ and its radiative forcing of $\Delta R_{[CO_2]}$ based on ice cores[37], Allan Hills Blue Ice[34], various marine proxies ($\delta^{11}B$ from either *T sacculifer*[10,12,57] or *G ruber*[11,58–61], alkenones[58,62,63]) paleosol[33], model simulation[9] and data-based reconstruction[38,39]. Error bars show $1\sigma$ uncertainty.

dynamics, as proposed earlier[8,15]. The hypotheses of pressure-melting point dynamics in North American land ice as obtained from the ice-sheet distribution underlying this study here[6] and of regolith removal[8,9,15] as an explanation for the MPT are not mutually exclusive. They all point to ice-sheet dynamics as a dominant process responsible for the lengthening of the glacial cycles after the MPT, but they disagree in detail on how this is achieved.

Climate change during the Quaternary consists not only of glacial/interglacial transitions, but contains also a long-term trend towards more positive values in the LR04 benthic $\delta^{18}O$ stack[4]. Decomposing the $\delta^{18}O$ signal in temperature and ice volume showed[29] that this trend is mainly based on a change in ice volume over the MPT rather than on a change in the temperature. This interpretation of little long-term change in the temperature of glacial maxima across the MPT agrees with a

reconstruction[31] of ocean temperature and sea level, but disagrees with another study[32], which mainly found a decrease in the glacial temperature across the MPT.

Proxy-based reconstructions of atmospheric $CO_2$ concentrations[10–12] (Fig. 4d) with the exception of paleosols[33] show a transition towards lower glacial values with time. These findings of stable interglacial $CO_2$ across the MPT, but higher glacial $CO_2$ before the MPT, are supported by snapshots of ice core data at 0.95 and 1.5 Myr BP from the Allan Hills blue ice area in Antarctica[34]. The question remains if climate and/or $CO_2$ is the driver for this evolution, and therefore also potentially for the MPT. Both model- and proxy-based reconstructions of $CO_2$ that extend beyond the 800-kyr covered by precise ice core $CO_2$ records[35–37] still lack the required precision of either $CO_2$ or age to resolve the most likely changes in the $CO_2$ radiative forcing across the MPT. $CO_2$ reconstructions seemed to some degree to depend on the underlying proxy[12] (Fig. 4d). Simulated $CO_2$ shows specifically small glacial/interglacial amplitudes of only 20–40 ppm early in the Quaternary[9], not yet contained in individual proxy-data sets or in the model-based synthesis of previous approaches[38,39]. Simulation experiments[40] have shown that glacial/interglacial transitions during the late Quaternary also occur when $CO_2$ is kept fixed, although no lengthening of the glacial cycles has then been observed, pointing towards a central role of a carbon cycle feedback during the MPT.

Various studies agree that the rise in the glacial/interglacial $CO_2$ amplitude across the MPT was probably caused by a mixed contribution from ocean circulation and enhanced iron fertilisation of the marine biota influencing the biological carbon pumps[9,41–43]. However, in the reconstructions and the carbon cycle modelling, the resulting changes in atmospheric $CO_2$ appeared to act as feedbacks in response to changes in the physical climate system giving evidence that $CO_2$ might contribute to the extension of glacial cycles across the MPT, but might not be its initial trigger. One exception, which suggests that $CO_2$ might be a trigger of the MPT, is a scenario in which the gradual $CO_2$ decline of the Quaternary is caused by changes in volcanic outgassing[9]. However, tests within this study have shown that the transition from 41 to 100-kyr cyclicity also appears without such externally driven changes in $CO_2$, therefore confirming the role of $CO_2$ as a feedback for the transition. This illustrates, that it is not yet clear whether $CO_2$ acted as trigger or feedback during the glacial/interglacial evolution across the MPT.

As stated earlier, we understand our land ice distribution as a model-based interpretation of the LR04 benthic $\delta^{18}O$ stack. Therefore, any uncertainties or inaccuracies in LR04 $\delta^{18}O$ are also contained in our results. The most recent update of the benthic $\delta^{18}O$ stack[44] is for the Quaternary very similar to the LR04 $\delta^{18}O$ with the expection of a time period around 1.8–1.9 Myr BP covering MIS 65–71. Here, amplitude and timing in both stacks differ, potentially influencing our detection of interglacials during three obliquity cycles, including MIS 65, and 71, for which we found skipped terminations. Furthermore, certain processes in the 3-D ice-sheet models need to be parametrized, adding another layer of uncertainty to the results. From a systematic parameter sensitivity study, it has been estimated[29] that these model-based uncertainties for the volume of the North American and Eurasian ice sheets are on average $0.5 \times 10^{15}$ m³ ($1\sigma$), which represents a relative uncertainty of 2%.

Reconstruction-based evidence on ice-sheet extents for these periods is rare. In a recent compilation[45], the proxy- and model-based knowledge on northern hemisphere land ice through the Quaternary has been combined, highlighting also large gaps in our knowledge on times earlier in the Quaternary. However, the feature[46] of a so far unexplained[47] early Pleistocene glacial advance in North America as far south as 38° N dated around

2.5 Myr BP is not yet met by our results and cannot be reconciled with our present knowledge on climate change. Either we are missing some details on radiative forcing that has been active in the early Quaternary, or these data need to be reinterpreted, or ice sheets models used for simulating these events have been too simplistic. Note that in the context of our study, especially the minimum in land ice volume related to interglacials is of interest, which is even more weakly constrained by data than the maximum glacial extent. Furthermore, these findings[46] suggest, that the existence of regolith might have promoted the existence of widespread North American ice sheets in the early Quaternary, but the absence of similar glacial advances for the following 1 Myr implies the absence of regolith thereafter. Those implications make these empirical data[46] difficult to reconcile with the regolith removal hypothesis[8].

Due to a lack of further independent data on the northern hemispheric ice volume outside of Greenland, other results of the model-based deconvolution[29] of the LR04 $\delta^{18}O$ stack might alternatively be used to evaluate of our results. Thus, sea level and $\delta^{18}O$ of seawater ($\delta^{18}O_{sw}$) of different approaches[29,48,49] have been compiled (Fig. 4c, d). Be aware that these are indirect comparisons, since they say nothing about our target variable. Furthermore, both variables will not give any information on the detected relatively larger contribution of ice sheets in Antarctica and Greenland to $\delta^{18}O_{sw}$ in the early compared to the late Quaternary (Fig. 2c). The approach of sea-level reconstruction based on Mediterranean data[48] is independent of the LR04 $\delta^{18}O$ stack (Fig. 4c), and suggests interglacial sea level rise of up to 50 m relative to present in the Early Quaternary. This would imply a shrinking of Antarctic ice sheets by more than 50%. To our knowledge, this is not supported by independent evidences and it puts the magnitude of sea-level change implied into question. The stack[49] of $\delta^{18}O_{sw}$ (Fig. 4d) consists of three cores from the North Atlantic[50,51], North Pacific[49], and South Pacific[31]. These $\delta^{18}O_{sw}$ records are calculated residuals once a Mg/Ca-based temperature contribution has been substracted from the measured benthic $\delta^{18}O$. Thus, the still existing huge variability and uncertainty in the stack might indicate an undersampling for a truly global mean $\delta^{18}O_{sw}$ and/or proxy-specific problems of this paleothermometer. Once future efforts produce a stack of $\delta^{18}O_{sw}$ based on an order of magnitude more sediment cores, such data might become quantitatively of interest for an evaluation of our results.

The latitudinal integration of land ice volume distribution across the Quaternary (Fig. 2b) clearly shows that northern hemisphere land ice was mostly situated north of 65°N before the MPT, and shifted southwards over time. In detail, at glacial maximum, the latitude at which exactly half of the northern hemisphere ice-volume change is north of it (called 50% line) shifted from about 70°N before 1.8 Myr BP to 62°N at the Last Glacial Maximum (LGM). The southern edge of the ice sheets during peak glacial times—here represented by the 10%-line of northern hemisphere ice-volume extent—moved from a variable position between 55 and 63°N early in the Quaternary to a stable 50°N after the MPT. Combining these temporal patterns of land ice distribution with orbital theory properly within the framework of an energy budget scheme[30,52] would give us a quantification of the land ice albedo feedback ($\Delta R_{[LI]}$)—one of the dominant processes changing climate across the MPT. This information can be used to evaluate how important the choice of the latitude at which changes in orbital insolation are investigated in simplified approaches[19,24] actually is. Interestingly, it turns out that although land ice before the MPT is mainly located north of 65°N, a precise knowledge of incoming solar energy (and therefore of the investigated latitude) is of secondary importance for the calculation of land ice albedo feedbacks. In other words, for the energy budget changes in land ice distribution and not

incoming insolation (or orbital theory) are most important. This implies that any reasonable latitude at which one finds northern hemispheric land ice (e.g., 55–75°N) is a valid choice at which insolation changes are analyzed justifying in retrospective the idea for simple rules based on incoming insolation at 65°N.

Focusing the analysis of the Quaternary on the definition of interglacials reduces the variety of climate change to a binary point of view: a time period is either a glacial or an interglacial. The comparison of our results of the definition of interglacials with those obtained by T17 clearly showed that especially for times earlier in the Quaternary ambiguity exists, and the outcome of any such binary definition depends to a large extent on the chosen thresholds, their uncertainties and the individual records involved in the analysis, e.g., the benthic LR04 $\delta^{18}O$ stack. This raises the question if any such strict definition of interglacials (and an in-depth analysis of their temporal pattern) is a meaningful exercise. We propose, alternatively, that the gradual change of climate should be more in the focus of further research. Normalized time series of LR04 $\delta^{18}O$, the detrended $\delta^{18}O$ used in T17 and of our northern hemispheric ice volume outside of Greenland clearly differ early in the Quaternary, while they converge to similar dynamics later on (Fig. 5). Any binary index derived from any of both variables contains only an extrapolation of the extremes, but is due to the chosen (partly subjective) definitions prone to errors or arbitrariness. Our conclusion that glacial/interglacial patterns are different in detail when based on northern hemispheric land ice outside of Greenland (this study) compared to an approach based on LR04 $\delta^{18}O$ (T17) would still hold, even when looking on these gradual changes that avoid any thresholds.

Our analysis is based on only one model-based interpretation of a benthic $\delta^{18}O$ stack and might therefore in some details be method dependent. However, our general finding, independent of uncertainties and approaches, is the following: The glacial/interglacial amplitudes of northern hemisphere land ice variability outside of Greenland are much smaller prior to than after the MPT. In the corresponding benthic $\delta^{18}O$ stack this change in glacial/interglacial amplitude across the MPT is much weaker, since $\delta^{18}O$ also contains contributions from deep ocean temperature and ice volume in Antarctica and Greenland.

While regular obliquity-driven changes in benthic $\delta^{18}O$ certainly appear prior to the MPT, our findings challenge if minima in benthic $\delta^{18}O$ should be classified as interglacials in the traditional sense used so far in the literature, since related amplitudes in northern hemispheric land ice outside of Greenland are rather small before the MPT. Consequently, our approach would lead to a pattern of climate change during the Quaternary that might consist of irregularly appearing interglacials both prior to and after the MPT. While the latter has been proposed before by various studies due to skipped terminations[6,8,9,19,20,24], the proposed irregular pattern prior to the MPT is, to our knowledge, new. Hence, the so-called 41-kyr world prior to the MPT would not be a period with regularly appearing glacial terminations (or new interglacials), but be irregularly glaciated and deglaciated similar to the so-called 100-kyr world after the MPT. In detail, the fraction of realized new interglacials or terminations per obliquity cycle is 67%, 88% and 52% during the early, mid, and late Quaternary, respectively. Alternatively, we have to accept that the definition of interglacials might not be applicable straightforward to the whole Quaternary since this conclusion is solely based on our applied definition of interglacials—the absence of significant northern hemispheric land ice outside of Greenland, which has been suggested[27] to be suitable for at least the last 800 kyr, and which agrees with other approaches[20,24] for the last 1.6 Myr. Following our definitions the appearance of interglacials cleary shifts across the Quaternary, from an interglacial-dominated to glacial-dominated climate. Furthermore, any scheme that tries to define glacial and interglacial periods depends not only on the chosen variable for classification, but also largely on chosen thresholds and their uncertainties. Such a binary view on climate introduces unnecessary ambiguity in the analysis. We propose that less importance should be placed on such classifications, and more efforts should be made in understanding and improving the underlying physical mechanisms driving the climate changes as recorded in the geological data, including at least carbon cycle feedbacks and changes in land ice.

## Methods

**Land ice distribution.** We use land ice simulation output[29] consisting of an updated version of an earlier study[6]. While the initial study[6] was restricted to land

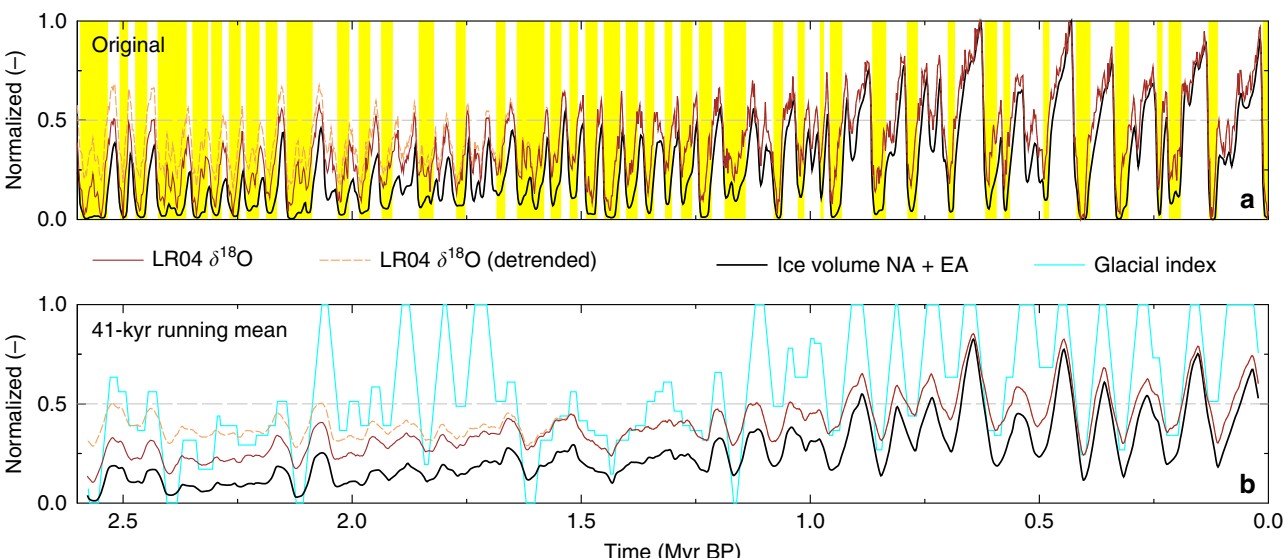

**Fig. 5 Indices of Quaternary climate. a** Original data including in yellow bars the identified interglacial periods (this study); **b** 41-kyr running means. Normalized time series of ice-volume change in North America (NA) and Eurasia (EA) used to define interglacials (this study) and of the LR04 benthic $\delta^{18}O$ stack[4] and its detrended version used in T17. In (**b**) the glacial index (the inverse to the yellow interglacial periods contained in (**a**)) is shown due to its in-phase dynamics to the other time series. This index is built, when the ice volume in North America and Eurasia is reduced to a binary pattern (glacial = 1; interglacial = 0).

ice dynamics in the northern hemisphere, the updated study now also simulates land ice in Antarctica. Within a modelling approach, the 3-D ice-sheet model ANICE is applied to deconvolve the benthic $\delta^{18}O$ stack LR04[4] in deep-ocean temperature, global mean sea-level variations, and a representation of the four main ice sheets in Antarctica, Greenland, Eurasia, and North America. In detail, the modelling framework is in a forward mode forced to match the temporal changes in reconstructed benthic $\delta^{18}O$ by physically consistent contributions from both deep ocean temperature change and land ice-volume (sea level) change. Areal extent and ice thickness of ice sheets and ice shelves from the four individually simulated ice masses in North America, Eurasia, Greenland, and Antarctica are available at their simulated spatial resolution of $40 \times 40$ km$^2$ (Greenland at $20 \times 20$ km$^2$). Results[29] cover the last 5 Myr, but we restrict our analysis to the Quaternary as done in T17. Temporal resolution of the available spatially resolved land ice changes is 2 kyr. For the detection of the onset of interglacials (see below), these data are interpolated to 1 kyr. This rather coarse temporal resolution of the ice-sheet model results, but also of the underlying benthic $\delta^{18}O$ stack LR04, does not allow us to test if a decrease in millennial-scale variability may have caused the lengthening of ice ages across the MPT, as suggested previously[26].

**Finding interglacials based on land ice volume**. The thresholds used to define interglacials have been chosen in a way to find similar results for the last 800 kyr as in a recent review paper[27] and to minimize the amount of obliquity cycles without interglacials earlier in time. Threshold 1 is the value below which the ice volume needs to fall to detect an interglacial, threshold 2 is the value above which the ice volume needs to rise to separate two interglacials from each other. If threshold 1 is not crossed within an obliquity cycle, this cycle is called to contain a *skipped termination* (in T17 also called *interstadial*). If threshold 2 is not crossed, the obliquity cycles contains a *continued interglacial*. In our standard case, the constraints given by the data and our goal to minimize the amount of obliquity cycles without interglacials lead to temporal changes in the chosen thresholds. For these, we use as additional boundary condition, that the difference between both thresholds stays constant over time. Being aware of uncertainties in the land ice volume of North American and Eurasian ice sheets, which have been estimated to $0.5 \times 10^{15}$ m$^3$ ($1\sigma$) in the model-based deconvolution of the LR04 benthic $\delta^{18}O$ stack[29], we choose threshold values only in full digits of $n_i \times 10^{15}$ m$^3$. In our standard case $n_1$ is 7 and $n_2$ 10 after 1650 kyr BP, and 3 and 6 before 1750 kyr BP, respectively, with linear interpolation inbetween. In the alternative definitions, the thresholds are kept constant over time, using either those values determined for the early or late Quaternary (Fig. 3b, c). Different to T17 (in which the onset of any interglacials needed to be closely related to a maxima in caloric summer half-year insolation), we here classify only if an obliquity cycle, defined by two following obliquity minima, contains the onset of a new interglacial irrespective of the precise timing. This should reduce the importance of chronological uncertainties earlier in the LR04 records on the outcome of our interglacial detection algorithm.

**Radiative forcing of $CO_2$**. The radiative forcing of $CO_2$ plotted in Fig. 4d is calculated after[53] $\Delta R_{[CO_2]} = 5.35$ W m$^{-2} \cdot \ln (CO_2/(278$ ppm$))$. The results of this formulation for typical Quaternary $CO_2$ values are close to those obtained with a revised approach[54], as has been shown elsewhere[55].

**Frequency analysis**. Frequency analyses have been performed with Analyseries[56].

## Data availability

Data taken from the cited literature are available via the referenced sources: LR04[4] benthic $\delta^{18}O$ (Figs. 1b, 4b, and 5); deconvolution of LR04 benthic $\delta^{18}O$ into temperature and land ice components (Fig. 2d) and the temporal and spatial distribution of land ice[29]; sea-level reconstruction[29,48] (Fig. 4c); $\delta^{18}O_{sw}$[29,49] (Fig. 4d); $CO_2$ reconstructions[9–12,33,34,37,39,57–63] (Fig. 4e). All plotted data are available at Pangaea with https://doi.org/10.1594/PANGAEA.914483.

## Code availability

In addition to the frequency analysis performed with the software Analyseries[56], and Fortran routines[64] used to calculate orbital parameters (obliquity, insolation), self-made scripts have been used for data analysis, which can be obtained from the first author.

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

## Acknowledgements
This paper contributes to PACES-II, the Helmholtz Research Programme of AWI, and to Netherlands Earth System Science Center programme *Reading the past to project the future*, funded by the Netherlands Organisation for Scientific Research (NWO). We thank Andrea Bleyer for language editing.

## Author contributions
P.K. designed research, analyzed data, prepared figures and led the writing of the draft. R.S.W.v.d.W. contributed with discussions and insights on the land ice distribution.

## Funding

## Competing interests
The authors declare no competing interests.
