## [Peer Review File · Nature Communications]

Reviewers' comments:

Reviewer #1 (Remarks to the Author):

This paper extends the ideas presented in Tzedakis et al. 2017 (T17): A simple rule to determine which insolation cycles lead to interglacials. Rather than basing the rule only on the insolation as in T17, the maximum glacier volume is more important for determining climate change.

I have read the paper twice, and perhaps I miss the point, but the message is very obscure and unclear to me: The point is to establish a relationship between the forcing (insolation changes) and the response (climate changes/glacial cycles). But the global ice extend, which the authors previously obtained from an ice sheet model (with insolation forcing) (refs 6 and 25), must be on the response side of the equation. I can fully understand the reasoning in explaining the (non-linear) ice volume response to the insolation changes, but I simply do not understand what the authors want to achieve in this paper.

The discussion of caloric summer half year, summer solstice insolation, 65N latitude insolation etc. (insolation proxies) is not new, and in my view not so important, when ice sheet modelling based on actual annually varying insolation fields is used. Obviously, if we had a proxy of the global ice volume and the extend (and thereby the ice albedo radiative forcing), we would be better off than we are with only benthic $\delta^{18}O$. This is the motivation for the "deconvolution" (ref 6). Is the point now to make a simple rule in connecting the obtained ice albedo with the insolation proxies?

I am reluctant in giving negative recommendation regarding publication, since I might have missed the point of the paper, but I will recommend a rewriting in order to convey the message more clearly. It should especially be reconsidered how the information in the figures (especially Figs 5 and 6) can be communicated better. The figures are difficult for me to understand.

I am sorry not to be more constructive in my criticism at this point.

Reviewer #2 (Remarks to the Author):

This manuscript presents a new theory for the definition of interglacials over the Quaternary, building and critiquing on the previous study by Tzedakis et al. 2017. The authors combine orbital theory with modelled land ice distribution to define their interglacials which they argue is more robust than using the multi-responsive benthic $\delta^{18}O$ stack. They find that ice albedo (a function of volume and location of ice) is the dominant factor impacting on climate, in front of any orbital parameters.

The paper is interesting and of broad scope, thus making it suitable for the audience of Nature Communications, however I find that in its current form it is missing out on some key deliverables which would improve the study greatly. Overall I find the paper difficult to read due to some poorly phrased sentences and sections, and I believe the conclusions and figures could be significantly clarified.

I infer from the discussion that the ice distribution model has by far the biggest role in the results, yet I see no discussion of the uncertainty pertaining to this and how it would impact on the results (especially as they are not 100% complete in their identification). For example, Line 111 (and throughout): Where your analysis presents something like a skipped termination, even where it is better than the previous efforts, I think some discussion is warranted to say which precise peaks/periods are missed and why this might be.

Line 157: Is this deterioration of the definition to do with the smaller signal to noise ratio of the E. Pleistocene cycles or is it merely to do with a lack of quality record across this interval? Worth discussion is whether this difficulty precludes the analysis from being run any further back than it already is.

The role of CO₂ forcing in the last 800 kyrs is very clear, and regardless of its nature as trigger or feedback (or both) I think more discussion is warranted on this important aspect of radiative balance, there are now many more available and reliable CO₂ proxy records for large portions of this interval and it seems odd to leave out discussion of these when they may help provide and explanation of the remaining misidentifications.

Figure 3 is unclear and I am not clear what it is supposed to show on this scale. Can you clarify this and

make the figure caption more explanatory.

I would recommend a new section towards the end of the manuscript summarising exactly the outcomes of this new analysis, as well as what further directions are required in this subject, as currently the end of the manuscript seems very abrupt.

Minor points

Lines 16-20: I find this sentence very difficult to penetrate, could you rephrase it and split it into two.

Line 30: The high resolution CO₂ studies (Chalk et al and Dyez et al) demonstrate this point more firmly, they should be used here in addition to later on.

Line 42-45: These sentences are awkwardly phrased.

Lines 86-88: I'm not sure I understand this reasoning, where does the threshold Tau come from, refer to the methods, and why is a time-dependency more important than a standard value?

Line 175: It is my understanding that MIS 100 at 2.6 million had some of the largest aerial ice extent of the Pleistocene, despite a modest volume (Balco, G. and Rovey, C.W., 2010. Absolute chronology for major Pleistocene advances of the Laurentide Ice Sheet. *Geology*, 38(9), pp.795-798.). How well do we really know these figures for ice latitude in individual glacials?

Line 207: Surely we are STILL limited by this information, as it is the most poorly constrained and most important part of your formulation.

Line 227: This becomes circular, as surely ice is the key feedback for these processes.

Line 259: Poor grammar, please rephrase.

Line 272: This is a really important point and I think it should come earlier in the text, and with more detail and context. Is it a fully inverted process or does it also have an essence of forward modelling as well?

Line 288: I think this is an important point, and one that has been acknowledged by the SL community at large. $\delta^{18}O$ and LI both have their shortcomings when used in this way, but neither are likely to be linearly related to the implied forcing, which means it is likely up to the individual approaches of these studies rather than a blanket SL = bad, $\delta^{18}O$ = good.

The last sentence of the conclusions is also mudd

Revision of paper

November 4, 2019

Before we give a point-by-point response to the critics of the reviewer we have to clarify that in this revision the definition of interglacials had to be revised, since we found an error in our calculations. Briefly, while we wanted to add land ice volume in North America and Eurasia to each other to obtain the changes in northern hemispheric ice volume outside of Greenland, we instead added ice volume in North America to ice volume in North America by using the wrong column of data. We realized this error only when we plotted the new Figure 2b, in which the ratio of ice volume in North America + Eurasia over all ice volume is shown, since this ratio was in the wrong calculation reaching values above one (which does not make sense). For transparency we plot below the initial (wrong) and the new (correct) ice volume change in North America + Eurasia, where you see that differences are not so large. However, the main findings of our study (interglacial are irregularly placed in between 2.6–1.6 Myr BP) is not affected by this error, but which obliquity cycle in detail contains an interglacial is affected. It implies that the thresholds which had to be crossed to define interglacials had to be revised. All is now documented in detail in the methods section. We apologise for this error.

Response to Reviews

Reviewer #1:

This paper extends the ideas presented in Tzedakis et al. 2017 (T17): A simple rule to determine which insolation cycles lead to interglacials. Rather than basing the rule only on the insolation as in T17, the maximum glacier volume is more important for determining climate change.

Our reply: It is not correct, that T17 based their simple rule only on insolation. They also need information on changes in the length of a glacial cycle, which is in T17 based on benthic $\delta^{18}\text{O}$, from which they calculate the effective energy needed to come to a glacial/interglacial transition (or termination).

I have read the paper twice, and perhaps I miss the point, but the message is very obscure and unclear to me: The point is to establish a relationship between the forcing (insolation changes) and the response (climate changes/glacial cycles). But the global ice extent, which the authors previously obtained from an ice sheet model (with insolation forcing) (refs 6 and 25), must be on the response side of the equation. I can fully understand the reasoning in explaining the (non-linear) ice volume response to the insolation changes, but I simply do not understand what the authors want to achieve in this paper.

Our reply: We agree, that we have been too unclear in our message, since we also compared our results a lot with previous studies. The question we like to answer is: Can we also predict the appearance of interglacials (similarly as in T17), if interglacials are defined by the disappearing of northern hemispheric land ice outside of Greenland (which is the original definition for interglacials) instead of only using benthic $\delta^{18}\text{O}$ as proxy for climate change? And the message we gain is: No. And we explain now in detail why this is the case (glacial/interglacial amplitudes in benthic $\delta^{18}\text{O}$ are smaller than glacial/interglacial amplitudes in northern hemispheric ice volume changes earlier in the Quaternary due to contribution of temperature and Antarctic ice volume changes to benthic $\delta^{18}\text{O}$; this in-depth analysis of the underlying causes for the different behaviour of benthic $\delta^{18}\text{O}$ and northern hemispheric ice volume was missing in our previously submitted version). We find, that interglacials are only regularly placed (one for every obliquity cycle) in the Mid Pleistocene, while they appear irregularly later-on (mainly due to too much ice and therefore skipped termination, which

was already known), but also before. This later finding is the new contribution, which is based on too long, continued interglacials. It implies that prior to the Mid-Pleistocene Transitions in the so-called 41-kyr world we also find an irregular pattern of interglacials. While irregularities in interglacials have already been suggested to take place after the Mid-Pleistocene Transitions (in the so-called 100-kyr world) due to skipped terminations, we now propose that throughout the Quaternary interglacials are irregularly placed. This is insofar new, that the regularity of glacial cycles in the 41-kyr world have by some authors been proposed as the "normal" obliquity-based circularity of the Quaternary, which has been perturbed by too long glacial periods during the last 1 Myr. Our new identification of interglacials would imply that also prior to the existence of large ice sheets in North America (happened to be the reason for skipped terminations during the last 1 Myr) climate change was rather irregular. The fundamental reasons of this new irregularity are not yet understood, it might be connected with changes in the greenhouse gases, but available data are too uncertain to come to conclusions.

The discussion of caloric summer half year, summer solstice insolation, 65N latitude insolation etc. (insolation proxies) is not new, and in my view not so important, when ice sheet modelling based on actual annually varying insolation fields is used.

Our reply: We also think that the discussion of different insolation metrics is of secondary importance, but we nevertheless think we have to show the difference when coming from caloric half-year insolation (used in T17) to full year insolation (used here). However, we have now shifted most of these details to supplementary figures and text found in the methods section, and only briefly touch on it in the main text.

Obviously, if we had a proxy of the global ice volume and the extend (and thereby the ice albedo radiative forcing), we would be better off than we are with only benthic $\delta^{18}\text{O}$. This is the motivation for the deconvolution (ref 6). Is the point now to make a simple rule in connecting the obtained ice albedo with the insolation proxies?

Our reply: As said already above our main point is to test what kind of climate pattern we gain across the Quaternary when the definition of interglacials is based on the absence of substantial northern hemispheric land ice volume outside of Greenland, since this definition has been shown to be objective for the last 800 kyr (Past Interglacials Working Group of PAGES, 2016), and is underlying the reasoning of T17, although they use LR04 benthic $\delta^{18}\text{O}$ in the end. The following details on ice albedo feedbacks were added to answer the two questions partly related to that (but of minor importance): Looking at land ice distribution and their latitudinal southward shift over time, how relevant is information on insolation obtained only at a fixed latitude? Knowing that the ice albedo feedback is the most important effect, can this be approximated by some simplifications? Both aspects led to two subsections in the results, which give more depth to the understanding, but which are not important for the main finding. We have already reduced the content of those subsections and the figures in the main text. If necessary, they can be shifted even completely to supplementary material or might even be deleted.

I am reluctant in giving negative recommendation regarding publication, since I might have missed the point of the paper, but I will recommend a rewriting in order to convey the message more clearly. It should especially be reconsidered how the information in the figures (especially Figs 5 and 6) can be communicated better. The figures are difficult for me to understand.

I am sorry not to be more constructive in my criticism at this point.

Our reply: The main finding is now much stronger highlighted and discussed leading to changes in title, abstracts and a new conclusion section. The figures have been revised, simplified and captions carefully extended. Part of the figures have been moved to the supplement, since some are not important for the main storyline. We also include now some new figures to highlight our findings, e.g. plotting CO_2 data along with our new land ice distribution-based definition of interglacials in figure 4. We hope this version of the draft will now be comprehensible to the readers including to easily see what the new aspect of this paper to the discussion of Quaternary climate change is.

Reviewer #2:

This manuscript presents a new theory for the definition of interglacials over the Quaternary, building and critiquing on the previous study by Tzedakis et al. 2017. The authors combine orbital theory with modelled land ice distribution to define their interglacials which they argue is more robust than using the multi-responsive benthic $\delta^{18}\text{O}$ stack. They find that ice albedo (a function of volume and location of ice) is the dominant factor impacting on climate, in front of any orbital parameters.

The paper is interesting and of broad scope, thus making it suitable for the audience of Nature Communications, however I find that in its current form it is missing out on some key deliverables which would improve the study greatly. Overall I find the paper difficult to read due to some poorly phrased sentences and sections, and I believe the conclusions and figures could be significantly clarified.

Our reply: We thank the reviewer for his efforts and for judging the paper suitable for this journal. We revised the paper significantly to clarify its messages. While in the initial version we had a lot of emphasis on how our results compare with previous studies, we now put those details upfront, which contain the new findings, that is that not only during the last 1 Myr interglacials appear irregular, but also during 2.6-1.6 Myr BP. Thus, the period before the Mid-Pleistocene Transition is according to our results not a time window with regularly (41-kyr periodicity) glacial cycles caused by obliquity, but also a time of irregularity. The reasons for that might need to be understood in detail in further studies. This major revision of our draft includes a new title and the focus on the most important findings. Some results are thus not shown anymore (formerly Fig 5d which compared the influence of different insolation metrics on the calculated land ice albedo feedback), or have been moved to the Supplements (formerly Fig 6 which shows how surrogates based on LR04 $\delta^{18}\text{O}$ or sea level compare with the full energy budget in the calculation of the land ice albedo feedback). We have now also included a new final figure, plotting our final classification of obliquity cycles with/without interglacials together with land ice changes in the northern hemisphere outside of Greenland, LR04 $\delta^{18}\text{O}$ and CO_2 reconstruction. In the methods we precisely lay out how interglacials are detected here based on northern hemispheric ice volume outside Greenland. Furthermore, figures have been clarified by simplification and revised captions.

I infer from the discussion that the ice distribution model has by far the biggest role in the results, yet I see no discussion of the uncertainty pertaining to this and how it would impact on the results (especially as they are not 100% complete in their identification). For example, Line 111 (and throughout): Where your analysis presents something like a skipped termination, even where it is better than the previous efforts, I think some discussion is warranted to say which precise peaks/periods are missed and why this might be.

Our reply: We extended the discussion on uncertainties of the modelled ice distribution, and also on details for those periods where we find different results than others. Briefly, we find eight obliquity periods with no new interglacials between 2.6 and 1.6 Myr BP, where others detect new interglacials. This difference in detecting interglacials can be understood by looking how amplitudes in benthic $\delta^{18}\text{O}$ are transferred into amplitudes in land ice distribution.

Line 157: Is this deterioration of the definition to do with the smaller signal to noise ratio of the E. Pleistocene cycles or is it merely to do with a lack of quality record across this interval? Worth discussion is whether this difficulty precludes the analysis from being run any further back than it already is.

Our reply: As mentioned in the reply to the previous comment, the difference in the classification of interglacials here and elsewhere depends on the investigated variable. Relatively amplitudes in benthic $\delta^{18}\text{O}$ are prior to the MPT mainly caused by temperature changes, and have also some contribution from ice volume changes in Antarctica and Greenland, making the residual amplitude caused by ice volume changes in North America and Eurasia to be used for interglacial detection here rather small. Signal-to-noise ratio does not seem to be an important aspect. So far, our approach has not been extended to older times, simply because we choose to focus on the Quaternary. But looking for similar things in the Pliocene might be a possibility.

The role of CO₂ forcing in the last 800 kyrs is very clear, and regardless of its nature as trigger or feedback (or both) I think more discussion is warranted on this important aspect of radiative balance, there are now many more available and reliable CO₂ proxy records for large portions of this interval and it seems odd to leave out discussion of these when they may help provide an explanation of the remaining misidentifications.

Our reply: Following this idea we have now included all available CO₂ data in a figure. We have also extended the discussion on the potential role of CO₂, but as laid out earlier, we understand that the difference between our interglacial detection and others is based on the investigated variable. However, CO₂ might play a role when explaining why some obliquity cycles have rather small amplitudes ($\delta^{18}\text{O}$ and land ice volume), but available CO₂ reconstructions are due to uncertainties not yet able to help here.

Figure 3 is unclear and I am not clear what it is supposed to show on this scale. Can you clarify this and make the figure caption more explanatory. I would recommend a new section towards the end of the manuscript summarising exactly the outcomes of this new analysis, as well as what further directions are required in this subject, as currently the end of the manuscript seems very abrupt.

Our reply: The mentioned figure contained frequency analysis of different insolation times series. We have revised the caption and enlarged the figure, but most importantly, this figure has been shifted to the Supplementary Figures, since it is not of major importance for our findings, but adds only some details to the discussion.

Minor points:

- 2.1. Lines 16-20: I find this sentence very difficult to penetrate, could you rephrase it and split it into two.

Our reply: The sentence has been simplified and sharpened for clarification

- 2.2. Line 30: The high resolution CO₂ studies (Chalk et al and Dyez et al) demonstrate this point more firmly, they should be used here in addition to later on.

Our reply: The mentioned studies (Chalk et al., 2017; Dyez et al., 2018) have been inserted here. Note, that the discussion of the role of CO₂ has been extended later in the manuscript, including a new figure showing available CO₂ data. Also note, that a recently published CO₂ data set based on paleosol across the Pleistocene (Da et al., 2019) seems to disagree with most other CO₂ reconstructions, making the role of CO₂ on these changes happening across the Mid-Pleistocene Transition even more uncertain.

- 2.3. Line 42-45: These sentences are awkwardly phrased.

Our reply: We rephrased the sentences for clarity.

- 2.4. Lines 86-88: I'm not sure I understand this reasoning, where does the threshold τ come from, refer to the methods, and why is a time-dependency more important than a standard value?

Our reply: The threshold τ the reviewer is referring to is introduced in Huybers (2006), but never in a time-dependent fashion. We clarify the text accordingly. Furthermore, we delete the symbol τ here, because it is only used twice in this paragraph and not in any other place in the manuscript. We are now more precise while this time-dependency in the threshold would be necessary, if it should be really used to find days with temperatures above freezing (which was the motivation in Huybers (2006) to use such a threshold for the definition of integrated summer insolation). This revision of the text also includes another reference to temperature reconstructions on Greenland based on bore hole measurements (Dahl-Jensen et al., 1998).

- 2.5. Line 175: It is my understanding that MIS 100 at 2.6 million had some of the largest aerial ice extent of the Pleistocene, despite a modest volume (Balco, G. and Rovey, C.W., 2010. Absolute chronology for major Pleistocene advances of the Laurentide Ice Sheet. *Geology*, 38(9), pp.795-798.). How well do we really know these figures for ice latitude in individual glacials?

Our reply: Our plotted land ice distributions are model-based deconvolution of the LR04 benthic $\delta^{18}\text{O}$ which especially takes care of physically consistent changes in $\delta^{18}\text{O}$ based on either temperature or sea level (land ice volume). Since this approach uses 3-D ice sheet models

the simulated changes in land ice are for itself also consistent with the current understanding of ice sheet physics. The spatial distribution of the ice sheets might indeed in detail be partly uncertain, which is one reason why we analyse only latitudinal bands in land ice, which cover the dominant effects of ice sheet evolution: its latitudinal position as function of time. We now acknowledge the data-based southwards extent published in the mentioned paper of Balco and Rovey (2010) in our discussion. Shakun (2017) also discusses these ice extents around 2.5 Myr and to explain them compares the ice sheet albedo forcing from three different approaches based on (i) sea level, (ii) on $\delta^{18}\text{O}$, and on (iii) an energy budget based on land ice albedo (our approach). From these three calculations the one similar to our approach can explain most of the necessary cooling. While all three approaches still fail to meet the necessary target to fully explain the extent of the ice sheets, our approach seemed at least to be the best of the available ideas so far. The comparison of these three approaches is also, what at the end of of manuscript is discussed. In this context it is helpful to understand that the land ice albedo radiative forcing seems indeed to be a valid simplification of the whole effect of land ice sheets (including the change in orography) on climate, as has been supported by model experiments (Roberts and Valdes, 2017). Interestingly, the findings of Balco and Rovey (2010) suggest, that the importance of regolith removal for North American ice sheets is only relevant briefly for MIS 96-100 around 2.5 Myr, but not in the 1 Myr between 2.5 and 1.5 Myr. This implication of Balco and Rovey (2010) would disagree with the most recent study of simulated climate and ice sheet evolution across the Quaternary by Willeit et al. (2019), which also does not simulate ice sheets in North America around 2.5 Myr as far south as suggested in Balco and Rovey (2010). For our paper here, however, the most important information for the definition of interglacials is the minimum extent of land ice in North America and Eurasia, not the maximum extent. We therefore believe, that not matching the data-based reconstruction of the southern most extent of the ice sheets around 2.5 Myr makes our approach not better (or worse) than others, but illustrates that there are still some knowledge gaps in our full understanding of ice sheet dynamics. These details are now taken up in the discussion section.

- 2.6. Line 207: Surely we are STILL limited by this information, as it is the most poorly constrained and most important part of your formulation.

Our reply: This comment refers to our statement, that Milankovitch was restricted in his interpretation by a lack of precise knowledge of changes in land ice distribution. We agree, that this information is still limited. However, the knowledge on land ice we nowadays have — based on marine $\delta^{18}\text{O}$ and physical 3-D land ice models, and further data-based evidence on ice sheet extent — is much more solid than has been in 1941, at the time of the publication of the book of Milankovitch cited here. Nevertheless, we revise this sentence carefully to clarify that there are still gaps in our knowledge of land ice evolution. Interestingly, recently an overview of our knowledge on land ice extent has been published in Nature Communications (Batchelor et al., 2019). This review of the state of the art combines both paleodata-based reconstructions and modelling, and will now also briefly be discussed in our manuscript.

- 2.7. Line 227: This becomes circular, as surely ice is the key feedback for these processes.

Our reply: The comment refers to our conclusion that for the energy budget not orbital theory, but changes in land ice distribution are most important. We can not see, how here a circular reasoning is involved: changes in the energy budget (changes in land ice albedo radiative forcing) has two inputs: (a) changes in incoming top-of-the-atmosphere insolation (what we refer to her as orbital theory) and (b) changes in land ice distribution. The idea of this section was to test if the very often taken choice of looking at incoming insolation at only one latitude (mostly 65°N) might lead to biased results. And we found that this is probably not the case, since the dominant effect of land ice on climate (which is the land ice albedo radiative forcing) depends a lot more on the ice distribution than on the incoming insolation.

- 2.8. Line 259: Poor grammar, please rephrase.

Our reply: This paragraph is now deleted due to word restrictions, since it does not add to important details.

- 2.9. Line 272: This is a really important point and I think it should come earlier in the text, and with more detail and context. Is it a fully inverted process or does it also have an essence of

forward modelling as well?

Our reply: This information on our approach being inverse is now contained in the introduction, when we introduce what we want to do. This is not a fully inverted approach. It uses reconstructed changes in LR04 $\delta^{18}\text{O}$ as driver for the simulated changes in $\delta^{18}\text{O}$ in the climate-ice sheet system, which is then physical consistent deconvolved into its temperature and sea level (ice sheet) contributions. More details are now given in the methods section.

- 2.10. Line 288: I think this is an important point, and one that has been acknowledged by the SL community at large. $\delta^{18}\text{O}$ and LI both have their shortcomings when used in this way, but neither are likely to be linearly related to the implied forcing, which means it is likely up to the individual approaches of these studies rather than a blanket SL = bad, $\delta^{18}\text{O}$ = good.

Our reply: We revised the sentence. We understand, that our way of calculating those surrogates by linear interpolations between two extreme cases (present day and LGM ice extent) is only one of many ways how this might be done - although probably the one applied frequently. It might also be the case that our energy budget itself is wrong. It is now also mentioned in the methods that the energy budget is a simplification with certain weaknesses, but nevertheless it was shown (Roberts and Valdes, 2017) that the full effect of land ice on climate is nearly identical to the albedo effect, while the effect of orography on climate can be neglected.

- 2.11. The last sentence of the conclusions is also mudd

Our reply: The whole conclusions has been revised, and this sentence does not appear anymore.

*References

- Balco, G. and Rovey, Charles W., I.: Absolute chronology for major Pleistocene advances of the Laurentide Ice Sheet, *Geology*, 38, 795–798, doi:10.1130/G30946.1, 2010.
- Batchelor, C. L., Margold, M., Krapp, M., Murton, D. K., Dalton, A. S., Gibbard, P. L., Stokes, C. R., Murton, J. B., and Manica, A.: The configuration of Northern Hemisphere ice sheets through the Quaternary, *Nature Communications*, 10, 3713, doi:10.1038/s41467-019-11601-2, 2019.
- Chalk, T. B., Hain, M. P., Foster, G. L., Rohling, E. J., Sexton, P. F., Badger, M. P. S., Cherry, S. G., Hasenfratz, A. P., Haug, G. H., Jaccard, S. L., Martínez-García, A., Pälike, H., Pancost, R. D., and Wilson, P. A.: Causes of ice age intensification across the Mid-Pleistocene Transition, *Proceedings of the National Academy of Sciences*, 114, 13 114–13 119, doi:10.1073/pnas.1702143114, 2017.
- Da, J., Zhang, Y. G., Li, G., Meng, X., and Ji, J.: Low CO_2 levels of the entire Pleistocene epoch, *Nature Communications*, 10, 4342, doi:10.1038/s41467-019-12357-5, 2019.
- Dahl-Jensen, D., Mosegaard, K., Gundestrup, N., Clow, G. D., Johnsen, S. J., Hansen, A. W., and Balling, N.: Past temperatures directly from the Greenland ice sheet, *Science*, 282, 268–271, 1998.
- Dyez, K. A., Hönisch, B., and Schmidt, G. A.: Early Pleistocene Obliquity-Scale pCO_2 Variability at ~ 1.5 Million Years Ago, *Paleoceanography and Paleoclimatology*, 33, 1270–1291, doi: 10.1029/2018PA003349, 2018.
- Huybers, P.: Early Pleistocene glacial cycles and the integrated summer insolation forcing, *Science*, 313, 508–511, doi:10.1126/science.1125249, 2006.
- Past Interglacials Working Group of PAGES: Interglacials of the last 800,000 years, *Reviews of Geophysics*, 54, 162–219, doi:10.1002/2015RG000482, 2016.
- Roberts, W. H. G. and Valdes, P. J.: Green Mountains and White Plains: The Effect of Northern Hemisphere Ice Sheets on the Global Energy Budget, *Journal of Climate*, 30, 3887–3905, doi: 10.1175/JCLI-D-15-0846.1, 2017.
- Shakun, J. D.: Modest global-scale cooling despite extensive early Pleistocene ice sheets, *Quaternary Science Reviews*, 165, 25 – 30, doi:https://doi.org/10.1016/j.quascirev.2017.04.010, 2017.
- Willeit, M., Ganopolski, A., Calov, R., and Brovkin, V.: Mid-Pleistocene transition in glacial cycles explained by declining CO_2 and regolith removal, *Science Advances*, 5, eaav7337, doi: 10.1126/sciadv.aav7337, 2019.

Reviewers' comments:

Reviewer #1 (Remarks to the Author):

In this second revision the authors make a fair attempt to accommodate to the questions and criticism raised by myself and the other reviewer. However, the paper and the points of the paper is still obscure to me. The authors write in their response:

The question we like to answer is: Can we also predict the appearance of interglacials (similarly as in T17), if interglacials are defined by the disappearing of northern hemispheric land ice outside of Greenland (which is the original definition for interglacials) instead of only using benthic $\delta^{18}O$ as proxy for climate change? And the message we gain is: No.

The authors find that in the early Pleistocene there are irregularities in the occurrence of interglacials as a function of the obliquity paced astronomical forcing. This is of course an interesting finding, but as I see it, the essential science is in deriving the (indirect) record of Northern Hemisphere ice volume/extend using the ice sheet model from the $\delta^{18}O$ benthic record and the (well known) insolation fields. Part of that is already reported in Ref 28, but if anything improved, that should be the main finding of the paper.

I'm not a geologist, thus not much concerned with defining "interglacials" as periods of no land ice -except from Greenland- in the Northern Hemisphere. That seems fine, but as the authors also state, there are quite some uncertainties in this measure, especially since minimum ice extend is "overwritten" by following ice advance. This means that as more and more geological evidence becomes available, the categorizations as interglacial periods can change. To me it seems more practical to just define the interglacials directly from the $\delta^{18}O$ record, which does not change -thus the definition becomes model-independent, though it might be that all interglacials are not the same with respect to actual land ice extents, which might be better reconstructed in the future.

Though I recommended otherwise, a substantial part of the manuscript is still discussing the insolation. Now with a term, which I do not understand: "insolation proxies". From Laskar's calculations, the insolation fields are accurately known. So, the only discussion relevant is for low resolution models (in time and/or space) of the ice sheets how the insolation fields governing the melt-off is parametrized (forgetting the -as important- accumulation (precipitation) field). It thus seems to me that there is a confusion of "proxy" and "parametrization"/"model dependent representation" of the insolation. (I couldn't make sense of the sentence on line 95).

To the benefit of the authors, I admit that I might miss some points here, so another interpretation of "insolation proxy" could be "proxy for occurrence of an interglacial", which Figure 1 seems to point at. The figure is difficult to read, (coloured vertical bars are grey?). Perhaps it would help with a closeup around the skipped termination at 2.1 MyrBP, together with a discussion on dating uncertainty and the difference between $\delta^{18}O$ and reconstructed land ice volume.

Lines 153ff, 251ff, 344ff: the deconvolution of the benthic $\delta^{18}O$ signal (ref 28) seems, again to me, to be the essential take home message (but what is new to this paper?).

I am sorry to say, that I am still reluctant in recommending publication, but I acknowledge, that there might be points that I'm simply missing.

Reviewer #2 (Remarks to the Author):

Review of 'Land ice distribution suggests an irregular pattern of interglacials across most of the Quaternary.'

I am re-reviewing this manuscript and so received the updated manuscript files as well as the rebuttal letter.

All-in-all I think the authors have done a good job in responding to the comments from myself and the other reviewer, and the new manuscript is much easier to read and more clearly communicative. However, I still feel like parts of the manuscript needs some clarification as it has taken me some time to fully (I hope) comprehend the meaning of some parts. I am now happy with the body of the science, and appreciate the honesty of the authors in correcting their previous mistake, I would never have picked it out! I have no doubt that the nature editorial and typesetting team will help with some of the more awkward phrasing, but I have highlighted some instances I feel are badly written below.

My largest remaining concern is the section 'Defining interglacials by land ice volume' as it contains many paragraphs which I had to read 2-5 times to work out their meaning. In general please makes sure that everywhere you are talking about exceptions and new definitions you make it very clear whether you are 'not finding something expected (by T17 or by LR04)' or 'finding something unexpected'. I wonder if the information presented here could be additionally summarised as a table with headers; MIS, age, classification, T17 classification, comment/notes.

I appreciate that the task of this paper is not to sort through and evaluate CO2 proxy data, but I find some of the discussion around these to be problematic. I think it is warranted to state that 'not all CO2 proxies' are equal in the discussion of agreement and disagreement. For example most boron (perhaps with the exception of ref 9) and ice core data agree well where they are coeval and this is summarised quite nicely in Yan et al. 2019. The statement 'CO2 reconstructions seemed to some degree depend on the underlying proxy12' where reference 12 compares 2 sites, 2 methods, 2 species and 2 distinct time periods all simultaneously cannot be accepted as a realistic test. Worse is that this point is reiterated more strongly in the conclusion where it is stated 'Available CO2 data are so far too sparse, too uncertain, and too much in disagreement with each other to fully exploit the role of CO2 radiative forcing for these dynamics.' Clearly more work needs to be done, but I think this statement undersells the huge achievement which has been made by this community. Please try to rephrase these discussions in light of these advances.

Lines 292-294: I think I alluded to this in my previous comments, but does this address the difference between ice volume and ice extent, given that thin, widespread ice will have a much larger radiative impact than thick localised ice, though they could appear the same in both $\delta^{18}O$ and SL?

Minor comments:

Line 14: Here, we define... This is a very run on sentence and requires splitting into at least 2. The first part 'as deduced for the last 800kyrs' sounds ambiguous without further contextualisation and I don't think is needed in the abstract.

Line 41: Remove 'anymore', comma after interglacials

Line 68: insert 'data rather than'

Line 76: Does surrogates here mean proxies? Or indices?

Line 95: This does not make sense to me as a sentence, please rephrase.

Line 107: about then a specified range is redundant.

Line 121: what about the potential bias from the stacking method and spatial heterogeneity.

Line 126: a glacial periods – either remove the a or the s.

Line 127-129: This sentence is confusing.

Lines 138-140: Here and with all sentences like this, please use more directional language to really clarify the point being made. i.e. is it a subsequent interglacial? Is MIS 33 connected to MIS 31 or MIS 35 by the 'failed' glacial. It takes heavy referencing of figures and exactly language to work out each of these cases.

Line 166: Please explain 'alternative negative interglacial detector'

Line 177: that is, they do not follow a simple rule that can be based on study of the LR04 stack with currently available information.

Line 183: Does it have to be just one latitude that is important?

Line 209: consists

Line 239: I cannot see the need for 'but'

Line 241: you state that any reasonable latitude is fine, and then just give the same example as is traditionally used. Could you state what reasonable is? (e.g. 40-80 N) ?

Line 252: a long term trend to more positive values.

Line 264: 'still not accurately enough' is incorrect, I would suggest that some of them are accurate enough, but perhaps lack the required precision of either CO₂ or age to be of definitive use here.

Line 283 uncertainties or inaccuracies in LR04.

Line 318: Mimic (no k)

Line 333: Do you have a handle on how stable these means are? It would be helpful to have an idea, such as from a +/- 2SD or similar measure. E.g. in the late Pleistocene they are typically spaced by ~80 or ~120 kyrs. Whereas the MPT sections really just has some double length IGs.

Line 362: I agree with this paragraph (until here) whole heartedly, clearly the 41kyr frequency is there, and in many proxy records, so I imagine we will be calling it the '41kyr world' for a while yet, but it reinforces that this does not necessarily mean true G and IG cycles as we know them now. Perhaps it does not need the following statements regarding CO₂ afterall.

Figure 1: Could you add the symbols into a legend?

Reviewer #3 (Remarks to the Author):

Let me start by saying that this a thought-provoking paper, with potentially important implications and as such, I suggest that it should be published in NComms, after some refocusing and revision, as explained below.

At the start of their response to Reviewer 1, Köhler and van de Wal (K&vdW) clearly articulate the aim of the paper: "Can we also predict the appearance of interglacials (similarly as in T17 [Tzedakis et al. 2017]), if interglacials are defined by the disappearing of northern hemispheric land ice outside of Greenland...". The go on to say that the answer is 'No', because interglacials were irregularly paced not only in the last 1 million years (Myr), but also before 1.6 Myr before present (Ma). While the skipping of insolation peaks leading to fewer deglaciations in the last 1 Myr is well known (and various hypotheses have been proposed to account for this), the authors suggest that the irregular pacing of interglacials before 1.6 Ma is new and undermines the idea of the so-called '41kyr-world' of the early Pleistocene. The implication is that the idea of early Pleistocene glacial-interglacial cycles as quasi-linear responses to obliquity cycles cannot be sustained in detail and therefore a simple rule cannot be applied.

There are two substantive issues with the MS in its present form: (i) the authors pursue too many different strands, which dilutes the focus of the paper; and (ii) while there is merit in the authors' observation that there is a difference between glacial cycles before and after 1.6 Myr, some of their specific proposals require adjustment.

K&vdW first apply the simple rule of T17 (an interglacial onset occurs when a peak in insolation exceeds a threshold that decreases with time elapsed since the previous deglaciation) by using a different insolation metric (full-year insolation [as opposed to caloric summer half-year insolation of T17] at 65°N and the LR04 benthic d18O stack. Their results are similar to those of T17, but with more misses. This exercise considers not only energy related to ablation, but also overall changes in the energy. However, as it is well known (and also shown in Fig. S1), the variance of full-year insolation is completely dominated by obliquity and that is why this metric maps very neatly on the grey bars of Fig. 1. But the problem is that sea-level determinations and glacial-geological evidence show that NH ice sheets also respond to precessional insolation variations (e.g. MIS 5e-d-c-b-a etc). The important point here is that any single insolation metric is simply a proportion of obliquity and precession, and the latitude and time interval we use is a particular way of describing that proportion. At 65°N, the variance of caloric summer insolation has almost equal obliquity and precession contributions and that is why it was used by T17. The authors correctly point out

that the latitude of interest should be where the ice was mostly situated, but the idea of a simple exercise is that a priori we don't know that.

K&vdW then consider the definition of interglacials. While T17 used the LR04 benthic $\delta^{18}O$ stack as an ice volume proxy, the authors, quite rightly, point out that the definition of interglacials as proposed by the PAGES Past Interglacials Working Group [PIGS] (2016) is the absence of NH ice outside Greenland. They propose therefore to use an inverse forward modelling approach (de Boer et al., 2014) to deconvolve the LR04 stack into deep-ocean temperature and global mean sea-level and eventually simulate changes in 4 ice sheets (N American, Eurasian, Greenland and Antarctica). They then use the obtained changes in N American ice sheet + Eurasian ice sheet to define interglacials by applying a higher and a lower threshold (similarly to T17) to distinguish interglacials from incomplete deglaciations and also from continued (or extended) interglacials, which follow skipped glacials. On this basis, they identify a number of skipped obliquity cycles in the last 1 Myr that are very similar to those of T17 (with possible differences in MIS 27 and 33). Several more differences with T17 emerge before 1.6 Ma, because the deconvolution of LR04 performed here suggests that changes in the benthic $\delta^{18}O$ were dominated by deep-water temperature variations and therefore ice volume changes were small. Unlike T17, K&vdW identify a series of interglacial onsets that were "missed" (MIS 55, 61, 65, 71, 77, 79, 93, 101). Of these, MIS 55, 79, 93, 101 are deemed to be continued interglacials because the preceding ice volume increase was too small to qualify as a glacial, while MIS 61, 65, 71 and 77 were deemed to be skipped interglacials because the residual ice volume was too large. The authors then suggest that with the exception of the period ~ 1.6 -1 Ma, most of the Quaternary was characterized by irregularly paced interglacials. They then propose that the PIGS 2016 definition of no significant ice outside Greenland might not be applicable to the entire Quaternary.

These are powerful suggestions that require closer examination. Inspection of the modelled NH ice volume in MIS 65 and 61 shows that they are extremely near misses (very near the threshold line). What they call MIS 77, actually looks like the missed interglacial at 2,062 Ma that was identified in T17, which was in line with weak caloric summer insolation below the deglaciation threshold (note however, that the sea level highstand for 2,062 does not look particularly different for those of MIS 53, 57, 59 in the Mediterranean sea level curve of Rohling et al, (2014; R14). MIS 71 looks like a truly skipped interglacial in the modelled NH ice volume of K&vdW. In T17, MIS 71 was classed as an uncertain interglacial; its caloric summer insolation was just below the deglaciation threshold, while its effective energy (incorporating a discount rate for the elapsed time) was just above it. On the other hand, examination of the Mediterranean sea level curve (R14) also suggests a normal interglacial highstand, but the $\delta^{18}O$ seawater of Sosdian and Rosenthal (2009; S&R09) indicates a weak interglacial.

In K&vdW, MIS 55, 79, 93 and 101 are considered continued interglacials because the modelled ice-volume increase preceding them was too small. In T17, the LR04 curve for the preceding cold stages MIS 56, 80, 94 and 102 just crossed the required threshold to qualify as glaciations, while in R14 and S&R09, the results are mixed. But the main point here is that the results of the modelled deconvolution suggest that what we have here is not missed interglacials, but missed glacials. The authors point out in their concluding section the predominance of interglacial conditions before 2.1 Ma and also between 1.7 and 1.1 Ma, but not during 1.7-2.1 Ma (Fig. 4a). Given the near misses and uncertainties regarding MIS 61, 65 and 71, I suggest that the picture that emerges is not one of missed interglacials but one of general interglacial dominance prior to 1.6 Ma.

With respect to the matter of the definition of interglacials, the first thing to say is that this is normally based on actual palaeodata, rather than model results. I understand that the results presented here use the LR04 benthic $\delta^{18}O$ stack as the initial input, but the output is the result of a modelling approach that translates LR04 into changing land ice distribution. So ideally, these results need to be compared with palaeodata. As far as I am aware, the only available palaeodata for the 1.6-2.6 Ma interval are the R2014 and S&R09 studies mentioned earlier, and these are not always in agreement with respect to the amplitude of the oscillations. I think the results of the modelling exercise presented here are extremely valuable, because of the potential implications, but verification would require a concerted effort from the palaeodata community in producing more deconvolved records before 1.5 Ma, in the way that it has been done recently by Ford et al. (2019 Geology).

Be that as it may, do we need a different definition of interglacials prior to 1.6 Ma? Here it is important to remember that the PIGS 2016 definition carries an uncertainty of ~20m sea-level equivalent. Thus the “no significant” ice outside Greenland in the definition is able to accommodate a wide range of situations. Inspection of the sea-level equivalent record of the combined North American and Eurasian ice sheets from de Boer et al. (2014) which is the basis for the current study, shows that in the interval 1.6-2.6 Ma all the highstands are above -20m s.l.e. Even if one were to apply stricter definitions, almost all of the highstands would qualify as interglacials, with the exception of the peak at 2.062 Ma and maybe MIS 71. Taken together, palaeodata and modelling results, therefore, suggest that of the 25 obliquity peaks between 2.6 and 1.6 Ma, 23 would be interglacials or continued interglacials. It seems to me that the simplest rule whereby interglacials occurred when caloric summer insolation boosted by high obliquity exceeded a simple deglaciation threshold still holds for the early Pleistocene, since continued interglacials are essentially neutral in this scheme (no glacial inception, therefore no deglaciation needed).

So where does this leave us with respect to the present MS? It has long been informally observed (on the basis of benthic isotope records) that “pure” 41kyr glacial cycles appear around the interval 1.6-1.25 Myr ago, while before that there are many instances where glacials were skipped, giving rise to continued interglacials. I would suggest that the important contribution of the present MS is that it draws attention to the fact that the early Pleistocene before 1 Ma is not one uniform interval. The discussions of insolation metrics and land ice albedo feedbacks may be interesting, but detract from the main focus of the study, which should be the small changes in NH ice volume before 1.6 Ma leading to skipped glacials.

In a pioneering modelling exercise over 20 years ago, in Berger et al. (1999, QSR) pointed out that “a striking feature of the simulated ice volume with a linearly decreasing CO₂ scenario over the past 3Ma is the reversal from a state of interglacial dominance in the early Pleistocene to glacial prevailing in late Pleistocene”. Clearly something changed in the climate system sometime around 1.6 Ma and I would suggest that the authors focus their discussion in that direction in a revised MS, rather than the suggestion that early Pleistocene should not be viewed as the 41-kyr world, which is not supported upon closer examination.

Chronis Tzedakis

Response to Reviews

Reviewer #1:

In this second revision the authors make a fair attempt to accommodate to the questions and criticism raised by myself and the other reviewer. However, the paper and the points of the paper is still obscure to me. The authors write in their response:

The question we like to answer is: Can we also predict the appearance of interglacials (similarly as in T17), if interglacials are defined by the disappearing of northern hemispheric land ice outside of Greenland (which is the original definition for interglacials) instead of only using benthic $\delta^{18}\text{O}$ as proxy for climate change? And the message we gain is: No.

The authors find that in the early Pleistocene there are irregularities in the occurrence of interglacials as a function of the obliquity paced astronomical forcing. This is of course an interesting finding, but as I see it, the essential science is in deriving the (indirect) record of Northern Hemisphere ice volume/extend using the ice sheet model from the $\delta^{18}\text{O}$ benthic record and the (well known) insolation fields. Part of that is already reported in Ref 28, but if anything improved, that should be the main finding of the paper.

Our reply: Indeed, with respect to ref 28 (de Boer et al., 2014), we here analyse the distribution of land ice as function of latitude and time, from which interglacials can be defined. This is the improvement on which we elaborate. According to this statement of the reviewer the core of our findings would be restricted to the model-based calculation of Northern Hemisphere ice volume out of LR04 $\delta^{18}\text{O}$, while to interpretation of that finding would be of little value. We believe this view on our study is too restricted for the following reason: For decades the scientific community has tried to understand why in the geological record we find 100-kyr periodicity during the last 1 Myr, while in the orbital insolation no dominant change on this frequency occurred. To take a look at the climate as a system that in principle only changes every 41-kyr due to obliquity and having a 100-kyr periodicity only as a consequence of skipped glacial terminations — first proposed by Huybers 2007, and supported with more in-depth analysis and a simple rule how interglacials are developing as function of insolation by Tzedakis et al 2017 (T17) — was a game changer in the understanding of Quaternary glacial-interglacial dynamics. We here contribute to this interpretation of T17 by testing if what they argue is the root cause of an interglacial (the absence of northern hemisphere land ice outside of Greenland) also follows the idea, that the Quaternary is in principle a 41-kyr regulated climate system with some exception during the last 1 Myr, and we find that this is not the case. To our understanding this finding is the core science of our approach, since it questions the earlier approaches and warrants us, that this view of regularly appear (every 41-kyr) interglacials might be probably too simple. Or in other words: Having the focus on the definition of interglacials and glacials reduces the climate of the Quaternary to a binary view, which to a certain extent depends on chosen thresholds for definition and the associated uncertainties. We now highlight in our revision the problem of this binary view a bit further in the discussion.

Im not a geologist, thus not much concerned with defining interglacials as periods of no land ice -except from Greenland- in the Northern Hemisphere. That seems fine, but as the authors also state, there are quite some uncertainties in this measure, especially since minimum ice extend is overwritten by following ice advance. This means that as more and more geological evidence becomes available, the categorizations as interglacial periods can change.

Our reply: The approaches taken here and in Tzedakis et al. (2017) on the definition of interglacials (and probably also various other definitions compiled in Past Interglacials Working Group of PAGES (2016)) depend on chosen thresholds, which have to be crossed by certain climate variables. We now in the end of our revised draft come to the conclusion, that all these approaches which rely on any thresholds extrapolate a gradually changing climate record into a binary index (glacial OR interglacial), which is prone to ambiguity.

To me it seems more practical to just define the interglacials directly from the d18O record, which does not change -thus the definition becomes model- independent, though it might be that all interglacials are not the same with respect to actual land ice extents, which might be better reconstructed in the future.

Our reply: We do not agree with the view, that $\delta^{18}\text{O}$ should be preferred above land ice distribution, since we have shown in Figure 2, that early in the Quaternary large peaks in $\delta^{18}\text{O}$ transform into rather small changes in northern hemispheric land ice volume. It is important to acknowledge that $\delta^{18}\text{O}$ and land ice behave differently early in the Quaternary, since the dominant effect of land ice on climate — the ice albedo feedback — is much smaller if based on model-based ice distributions than if based on $\delta^{18}\text{O}$ alone. If this difference is ignored, different conclusions might be drawn from the available time series. Also note, that in T17 the LR04 $\delta^{18}\text{O}$ has been detrended between 1.5 and 2.6 Myr BP before analysis to account for secondary effects of temperature and Antarctic ice volume. This has now been discussed in our paper in more details, since it is a rather rough correction, which in our view can be done better by applying the model-based deconvolution as done here.

Though I recommended otherwise, a substantial part of the manuscript is still discussing the insolation.

Our reply: The sections on insolation had already been shortened in the last iteration (e.g. a lot of the related figures had been moved to the SI figures), but they have now been removed from the draft in order to refocus.

Now with a term, which I do not understand: insolation proxies. From Laskar's calculations, the insolation fields are accurately known. So, the only discussion relevant is for low resolution models (in time and/or space) of the ice sheets how the insolation fields governing the melt-off is parametrized (forgetting the -as important- accumulation (precipitation) field). It thus seems to me that there is a confusion of proxy and parametrization/model dependent representation of the insolation. (I couldn't make sense of the sentence on line 95).

Our reply: If I counted correctly, the term “insolation proxy” has once been mentioned in our draft, and that was a false description of what has been meant, for which we have to apologize. What has been meant was “insolation metric”, which discusses, if fully year insolation, peak summer insolation, or summer half year insolation might be the relevant metric to be investigated here. Note, that this has been done in a similar way in Tzedakis et al 2017, and also note that the part on insolation metrics has been removed from the draft.

To the benefit of the authors, I admit that I might miss some points here, so another interpretation of insolation proxy could be proxy for occurrence of an interglacial, which Figure 1 seems to point at. The figure is difficult to read, (coloured vertical bars are grey?). Perhaps it would help with a closeup around the skipped termination at 2.1 MyrBP, together with a discussion on dating uncertainty and the difference between d18O and reconstructed land ice volume.

Our reply: As already said in the reply to the previous comment, we never meant to use the term “insolation proxy”. The insolation metrics plotted previously in Fig 1 are full year insolation (FYI) and caloric summer half-year insolation (CSI), and they have been both shown here in comparison, since in T17 it is developed how changes in insolation (in detail CSI) led to interglacials and our first effort was to reproduce those findings of T17. However, in the revised version all details on insolation are cut short and in Fig 1 FYI and CSI are not plotted anymore.

Lines 153ff, 251ff, 344ff: the deconvolution of the benthic d18O signal (ref 28) seems, again to me, to be the essential take home message (but what is new to this paper?).

Our reply: As already stated above we believe our interpretation of land ice distribution for the definition of interglacials, that can then be compared to T17, is the essential new finding of our draft which warrants the publication of this paper.

I am sorry to say, that I am still reluctant in recommending publication, but I acknowledge, that there might be points that I'm simply missing.

Our reply: We hope that the refocusing and thinning out of the paper helps to convince the reviewer that this article might now be recommended for publication.

Reviewer #2:

I am re-reviewing this manuscript and so received the updated manuscript files as well as the rebuttal letter. All-in-all I think the authors have done a good job in responding to the comments from myself and the other reviewer, and the new manuscript is much easier to read and more clearly communicative. However, I still feel like parts of the manuscript needs some clarification as it has taken me some time to fully (I hope) comprehend the meaning of some parts. I am now happy with the body of the science, and appreciate the honesty of the authors in correcting their previous mistake, I would never have picked it out!

I have no doubt that the nature editorial and typesetting team will help with some of the more awkward phrasing, but I have highlighted some instances I feel are badly written below.

My largest remaining concern is the section Defining interglacials by land ice volume as it contains many paragraphs which I had to read 2-5 times to work out their meaning. In general please makes sure that everywhere you are talking about exceptions and new definitions you make it very clear whether you are not finding something expected (by T17 or by LR04) or finding something unexpected. I wonder if the information presented here could be additionally summarised as a table with headers; MIS, age, classification, T17 classification, comment/notes.

Our reply: Since it had been a comment by all reviewers we have heavily revised, shortened and thinned out the draft in an effort to reduce it to the main message. The idea of a table is certainly of interest. However, from the discussion we developed below in response in the comments of reviewer #3 we believe that probably the opposite is true: This binary view on climate focusing on defining periods either as glacial or interglacial (as done in T17 and as done here) is a simplification that has certain drawbacks. This view might help us in communicating, but also has the pitfall of oversimplification included. Already the discussion on threshold and uncertainties of thresholds shows that there is certain ambiguity to this approach which might be overcome by looking more on the gradually evolving consequences of the climate system.

I appreciate that the task of this paper is not to sort through and evaluate CO₂ proxy data, but I find some of the discussion around these to be problematic. I think it is warranted to state that not all CO₂ proxies are equal in the discussion of agreement and disagreement. For example most boron (perhaps with the exception of ref 9) and ice core data agree well where they are coeval and this is summarised quite nicely in Yan et al. 2019. The statement "CO₂ reconstructions seemed to some degree depend on the underlying proxy (ref 12)" where reference 12 compares 2 sites, 2 methods, 2 species and 2 distinct time periods all simultaneously cannot be accepted as a realistic test. Worse is that this point is reiterated more strongly in the conclusion where it is stated "Available CO₂ data are so far too sparse, too uncertain, and too much in disagreement with each other to fully exploit the role of CO₂ radiative forcing for these dynamics." Clearly more work needs to be done, but I think this statement undersells the huge achievement which has been made by this community. Please try to rephrase these discussions in light of these advances.

Our reply: We carefully revised the discussion on CO₂ as suggested. Clearly, CO₂ is not the main focus of this paper, and we are happy to improve how, when, and where the different CO₂ proxies has been discussed. Also note, that we now also referred to the original papers in which CO₂ proxies have been published, and not only to the compilation of Dyez et al. (2018).

Lines 292-294: I think I alluded to this in my previous comments, but does this address the difference between ice volume and ice extent, given that thin, widespread ice will have a much larger radiative impact than thick localised ice, though they could appear the same in both $\delta^{18}\text{O}$ and SL?

Our reply: If for some reason the properties of the ice dynamics change over time (e.g. presence

of sediments below the ice), one would go from thick localised to thin widespread with the same ice volume, but the required temperature change to match the $\delta^{18}\text{O}$ will be smaller: the radiation term in the energy budget is more important because of the larger surface area. So the net effect will be a dampening of the marine $\delta^{18}\text{O}$ amplitudes whereas the amplitude in SL stays similar. So in summary we think that removal of sediment leads to changes in ice dynamics favoring an increase in the marine $\delta^{18}\text{O}$ amplitudes. Hence distinguishing the regolith hypothesis from the merging/sliding is difficult because they both work in the same direction in the marine benthic $\delta^{18}\text{O}$ record.

Minor comments:

Our reply: All these minor comments have been considered, however due to some major reorganisations of the draft, it might be that the relevant paragraphs are no longer part of the revised manuscript.

- Line 14: Here, we define... This is a very run on sentence and requires splitting into at least 2. The first part as deduced for the last 800kyrs sounds ambiguous without further contextualisation and I dont think is needed in the abstract.
- Line 41: Remove anymore, comma after interglacials
- Line 68: insert data rather than
- Line 76: Does surrogates here mean proxies? Or indices? **Our reply:** None of both, we reformulate “surrogate” as “simplified calculations”.
- Line 95: This does not make sense to me as a sentence, please rephrase.
- Line 107: about then a specified range is redundant. **Our reply:** Nothing done, since we did not understand what has been meant here.
- Line 121: what about the potential bias from the stacking method and spatial heterogeneity.
- Line 126: a glacial periods either remove the a or the s.
- Line 127-129: This sentence is confusing.
- Lines 138-140: Here and with all sentences like this, please use more directional language to really clarify the point being made. i.e. is it a subsequent interglacial? Is MIS 33 connected to MIS 31 or MIS 35 by the failed glacial. It takes heavy referencing of figures and exactly language to work out each of these cases. **Our reply:** The confusion might arise from our way of explaining our results: We first discuss the difference to T17, but only later-on explain in detail what happened in the MIS (e.g. is it a skipped termination or a continued interglacial). Adding more details (as given by reviewer #3) the explanation of what might have happened has been enhanced and put more upfront. Hopefully, this also satisfies the comment here.
- Line 166: Please explain alternative negative interglacial detector
- Line 177: that is, they do not follow a simple rule that can be based on study of the LR04 stack with currently available information.
- Line 183: Does it have to be just one latitude that is important?
- Line 209: consists
- Line 239: I cannot see the need for but
- Line 241: you state that any reasonable latitude is fine, and then just give the same example as is traditionally used. Could you state what reasonable is? (e.g. 40-80 N) ?
- Line 252: a long term trend to more positive values.
- Line 264: still not accurately enough is incorrect, I would suggest that some of them are accurate enough, but perhaps lack the required precision of either CO2 or age to be of definitive use here.
- Line 283 uncertainties or inaccuracies in LR04.

- Line 318: Mimic (no k)
- Line 333: Do you have a handle on how stable these means are? It would be helpful to have an idea, such as from a $\pm 2SD$ or similar measure. E.g. in the late Pleistocene they are typically spaced by 80 or 120 kyrs. Whereas the MPT sections really just has some double length IGs.
- Line 362: I agree with this paragraph (until here) whole heartedly, clearly the 41kyr frequency is there, and in many proxy records, so I imagine we will be calling it the 41kyr world for a while yet, but it reinforces that this does not necessarily mean true G and IG cycles as we know them now. Perhaps it does not need the following statements regarding CO2 afterall.
- Figure 1: Could you add the symbols into a legend?

Reviewer #3:

Let me start by saying that this a thought-provoking paper, with potentially important implications and as such, I suggest that it should be published in NComms, after some refocusing and revision, as explained below.

At the start of their response to Reviewer 1, Köhler and van de Wal (KvdW) clearly articulate the aim of the paper: Can we also predict the appearance of interglacials (similarly as in T17 [Tzedakis et al. 2017]), if interglacials are defined by the disappearing of northern hemispheric land ice outside of Greenland.... The go on to say that the answer is No, because interglacials were irregularly paced not only in the last 1 million years (Myr), but also before 1.6 Myr before present (Ma). While the skipping of insolation peaks leading to fewer deglaciations in the last 1 Myr is well known (and various hypotheses have been proposed to account for this), the authors suggest that the irregular pacing of interglacials before 1.6 Ma is new and undermines the idea of the so-called 41kyr-world of the early Pleistocene. The implication is that the idea of early Pleistocene glacial-interglacial cycles as quasi-linear responses to obliquity cycles cannot be sustained in detail and therefore a simple rule cannot be applied.

There are two substantive issues with the MS in its present form: (i) the authors pursue too many different strands, which dilutes the focus of the paper; and (ii) while there is merit in the authors observation that there is a difference between glacial cycles before and after 1.6 Myr, some of their specific proposals require adjustment.

Our reply:

To (i): The paper was refocused as suggested.

To (ii): The detailed discussion of our results and its potential consequences has been refined, using the suggestion of the reviewer given below.

KvdW first apply the simple rule of T17 (an interglacial onset occurs when a peak in insolation exceeds a threshold that decreases with time elapsed since the previous deglaciation) by using a different insolation metric (full-year insolation [as opposed to caloric summer half-year insolation of T17] at 65°N and the LR04 benthic d18O stack.

Their results are similar to those of T17, but with more misses. This exercise considers not only energy related to ablation, but also overall changes in the energy. However, as it is well known (and also shown in Fig. S1), the variance of full-year insolation is completely dominated by obliquity and that is why this metric maps very neatly on the grey bars of Fig. 1. But the problem is that sea-level determinations and glacial-geological evidence show that NH ice sheets also respond to precessional insolation variations (e.g. MIS 5e-d-c-b-a etc). The important point here is that any single insolation metric is simply a proportion of obliquity and precession, and the latitude and time interval we use is a particular way of describing that proportion. At 65°N, the variance of caloric summer insolation has almost equal obliquity and precession contributions and that is why it was used by T17. The authors correctly point out that the latitude of interest should be where the ice was mostly situated, but the idea of a simple exercise is that a priori we dont know that.

Our reply:

The intention of paper was not to show that the simple rule found in T17 (how an insolation cycle leads to an interglacial) is not working. Our intention is to get one step beyond the previous approach by not using benthic \$\delta^{18}\text{O}\$ as climate record to be analysed but the variable that has been identified by Past Interglacials Working Group of PAGES (2016) to work best as a criterion for interglacials for the last 800 kyr: the absence of northern hemispheric land ice outside of Greenland.

Our point that in principle the latitude of interest should be where we ice was situated was made as a cross-check on the widely used application of insolation at 65°N. However, our energy balance

investigations have shown, that no matter where the ice is, a detailed and correct pick of the latitude for insolation is of secondary importance since the ice albedo feedback is a lot larger than the initial perturbation in insolation and even with a weakly picked latitude (say 65°N early in the Quaternary, when ice is mainly around 75°N), calculated changes in the energy balance are still acceptable. We believe such insights are useful for the wider community. Therefore, this finding is still contained in one short paragraph in the discussion, even with the full result section on the energy balance analysis (including all figures) being removed in their revision. Furthermore, we argue that we should not ignore land ice distribution available from models nowadays, but should make the best out of available information, and not rely on simple models applied on a restricted set of available data.

KvdW then consider the definition of interglacials. While T17 used the LR04 benthic d18O stack as an ice volume proxy, the authors, quite rightly, point out that the definition of interglacials as proposed by the PAGES Past Interglacials Working Group [PIGS] (2016) is the absence of NH ice outside Greenland. They propose therefore to use an inverse forward modelling approach (de Boer et al., 2014) to deconvolve the LR04 stack into deep-ocean temperature and global mean sea-level and eventually simulate changes in 4 ice sheets (N American, Eurasian, Greenland and Antarctica). They then use the obtained changes in N American ice sheet + Eurasian ice sheet to define interglacials by applying a higher and a lower threshold (similarly to T17) to distinguish interglacials from incomplete deglaciations and also from continued (or extended) interglacials, which follow skipped glacials. On this basis, they identify a number of skipped obliquity cycles in the last 1 Myr that are very similar to those of T17 (with possible differences in MIS 27 and 33). Several more differences with T17 emerge before 1.6 Ma, because the deconvolution of LR04 performed here suggests that changes in the benthic d18O were dominated by deep-water temperature variations and therefore ice volume changes were small. Unlike T17, KvdW identify a series of interglacial onsets that were missed (MIS 55, 61, 65, 71, 77, 79, 93, 101). Of these, MIS 55, 79, 93, 101 are deemed to be continued interglacials because the preceding ice volume increase was too small to qualify as a glacial, while MIS 61, 65, 71 and 77 were deemed to be skipped interglacials because the residual ice volume was too large. The authors then suggest that with the exception of the period 1.6-1 Ma, most of the Quaternary was characterized by irregularly paced interglacials. They then propose that the PIGS 2016 definition of no significant ice outside Greenland might not be applicable to the entire Quaternary.

Our reply:

This is a nice summary of our findings. By reading it in the words of the reviewer we realized that our final conclusion (the last sentence above) might need a slight reformulation.

These are powerful suggestions that require closer examination. Inspection of the modelled NH ice volume in MIS 65 and 61 shows that they are extremely near misses (very near the threshold line). What they call MIS 77, actually looks like the missed interglacial at 2,062 Ma that was identified in T17, which was in line with weak caloric summer insolation below the deglaciation threshold (note however, that the sea level highstand for 2,062 does not look particularly different for those of MIS 53, 57, 59 in the Mediterranean sea level curve of Rohling et al, (2014; R14). MIS 71 looks like a truly skipped interglacial in the modelled NH ice volume of KvdW. In T17, MIS 71 was classed as an uncertain interglacial; its caloric summer insolation was just below the deglaciation threshold, while its effective energy (incorporating a discount rate for the elapsed time) was just above it. On the other hand, examination of the Mediterranean sea level curve (R14) also suggests a normal interglacial highstand, but the d18Oseawater of Sosdian and Rosenthal (2009; SR09) indicates a weak interglacial.

Our reply: Our in detail responses:

MIS 55 and 65 have already been detected as one of the obliquity cycles of last 2 Myr with a skipped termination (Huybers, 2007), although our analysis detects MIS 55 as a continued interglacial. MIS 61, 65: True, the NH ice sheet in these MIS is close to the threshold, but as seen in the sensitivity tests (former SI Fig 4) with different thresholds the amount of obliquity cycles without a

new interglacial is more likely to increase (not to decrease) when changing the threshold.

MIS 77: As seen from the yellow colour-code in Fig 2 we do not claim that we find something different here than T17, both studies agree in having an obliquity cycle without new interglacial here. The sea level curve of Rohling et al. (2014) is for interglacials between 1.5 and 2.6 Myr 30–50 m above present day (de Boer et al. (2014) underlying our study here find interglacial sea levels between 0 and -20 m), which would imply a shrinking of Antarctic ice sheets by more than 50%, which is to our knowledge not supported by independent evidences.

MIS 71: This is a nice example of the dependence on thresholds for the detection of interglacial, which supported our choice to be in the end of our draft rather critical to such classification schemes.

In KvdW, MIS 55, 79, 93 and 101 are considered continued interglacials because the modelled ice-volume increase preceding them was too small.

Our reply: Our applied criteria (following Past Interglacials Working Group of PAGES (2016)) is: there needs to be a glacial defined by a threshold in NH ice volume outside Greenland between two interglacials, otherwise its a long (continued) interglacials. This is not exactly the same as written here.

In T17, the LR04 curve for the preceding cold stages MIS 56, 80, 94 and 102 just crossed the required threshold to qualify as glaciations, while in R14 and SR09, the results are mixed. But the main point here is that the results of the modelled deconvolution suggest that what we have here is not missed interglacials, but missed glacials.

Our reply: To our understanding “missed glacials” as named by the reviewer is the same as “extended or continued interglacials” which has been used in T17, and therefore for reasons of consistency also be used in our draft.

The authors point out in their concluding section the predominance of interglacial conditions before 2.1 Ma and also between 1.7 and 1.1 Ma, but not during 1.7-2.1 Ma (Fig. 4a). Given the near misses and uncertainties regarding MIS 61, 65 and 71, I suggest that the picture that emerges is not one of missed interglacials but one of general interglacial dominance prior to 1.6 Ma.

Our reply: We clearly state in our conclusion, that we find between 2.6 and 1.6 Myr a mixture of 4 continued interglacials (called missed glacials by the reviewer) and of 4 skipped terminations. We therefore can not see why the reviewer argues we suggest “that the picture that emerges is not one of missed interglacials”. Whether one agrees with the notation of “general interglacial dominance prior to 1.6 Ma” depends in the end if one wants to distinguish a binary pattern of glacial OR interglacials in the climate record, which is suggested by the search for and definition of interglacials, or whether one understands the climate system as a system which gradually evolves. Reflecting again on the difficulties we have to overcome with to fulfil the binary view, we tend to think that maybe one needs to get away from this view, since climate itself does not care for, if I call a situation an interglacial or a glacial. Relevant for the energy budget (and therefore for climate) is mainly the size the related ice and ice-albedo feedback. We have extended the discussion in the revise draft in that direction.

With respect to the matter of the definition of interglacials, the first thing to say is that this is normally based on actual palaeodata, rather than model results. I understand that the results presented here use the LR04 benthic $\delta^{18}O$ stack as the initial input, but the output is the result of a modelling approach that translates LR04 into changing land ice distribution. So ideally, these results need to be compared with palaeodata. As far as I am aware, the only available palaeodata for the 1.6-2.6 Ma interval are the R2014 and SR09 studies mentioned earlier, and these are not always in agreement with respect to the amplitude of the oscillations. I think the results of the modelling

exercise presented here are extremely valuable, because of the potential implications, but verification would require a concerted effort from the palaeodata community in producing more deconvolved records before 1.5 Ma, in the way that it has been done recently by Ford et al. (2019 Geology).

Our reply: With respect to the statement that “the definition of interglacials ... is normally based on actual paleodata” we like to point out that LR04 $\delta^{18}\text{O}$ needed to be detrended before 1.5 Myr BP in order to fit in T17 in the desired analysis scheme. This detrending has been motivated by the effect of deep ocean temperature and Antarctic ice volume. In fact, it is a similar approach than in our study in which we have to accept a change in the chosen threshold (and not a change in the underlying data), but nevertheless it consists of a correction of the data for secondary effects, which we argue might have been performed in a more quantitative sense with the model-based deconvolution of LR04 $\delta^{18}\text{O}$ underlying our study. The details on this detrending have been missing so far in our draft, but have now been included in the revision.

We agree that evidence from different directions in support for a hypothesis is always good. However, the data-based papers mentioned above (Rohling et al., 2014; Ford and Raymo, 2019; Sosdian and Rosenthal, 2009; Elderfield et al., 2012) all follow a similar approach: They calculate temperature from a different proxy (Mg/Ca), subtract the temperature related $\delta^{18}\text{O}$ from the measured benthic $\delta^{18}\text{O}$ to come up with sea water $\delta^{18}\text{O}$ ($\delta^{18}\text{O}_{\text{sw}}$), typically caused by land ice. In R14 sea level is deduced from a different approach, from which $\delta^{18}\text{O}_{\text{sw}}$ is determined. In all approaches some kind of model is involved, e.g. the regression between Mg/Ca and temperature change. Since those so-called data-based approaches disagree quite a lot we can not see, that they should contain a solution that might be better or even preferable to the model-based deconvolution of $\delta^{18}\text{O}$ which has been performed in de Boer et al. (2014), at least here the obtained changes for sea level and temperature are physically consistent. It might be that these data-based approaches become better, once more than only a few records are considered (eg. Ford and Raymo (2019) stacked 3 records, but still has huge uncertainties) - similar as LR04 $\delta^{18}\text{O}$ is qualitatively a lot better than individual $\delta^{18}\text{O}$ records, it might also be that the Mg/Ca paleothermometer contains some inherent uncertainties, that will prevent any significant step change in the near future. All these issues are now discussed in the revised draft.

Be that as it may, do we need a different definition of interglacials prior to 1.6 Ma? Here it is important to remember that the PIGS 2016 definition carries an uncertainty of $\sim 20\text{m}$ sea-level equivalent. Thus the “no significant” ice outside Greenland in the definition is able to accommodate a wide range of situations. Inspection of the sea-level equivalent record of the combined North American and Eurasian ice sheets from de Boer et al. (2014) which is the basis for the current study, shows that in the interval 1.6-2.6 Ma all the highstands are above 20m s.l.e. Even if one were to apply stricter definitions, almost all of the highstands would qualify as interglacials, with the exception of the peak at 2.062 Ma and maybe MIS 71. Taken together, palaeodata and modelling results, therefore, suggest that of the 25 obliquity peaks between 2.6 and 1.6 Ma, 23 would be interglacials or continued interglacials. It seems to me that the simplest rule whereby interglacials occurred when caloric summer insolation boosted by high obliquity exceeded a simple deglaciation threshold still holds for the early Pleistocene, since continued interglacials are essentially neutral in this scheme (no glacial inception, therefore no deglaciation needed).

Our reply: Note, that we emphasis in Fig 2, that not only benthic $\delta^{18}\text{O}$, but also sea level is not via a constant factor related to NH ice volume outside Greenland, since the relative contribution of ice in Antarctica and Greenland to $\delta^{18}\text{O}_{\text{sw}}$ has been larger earlier in the Quaternary compared to the late Quaternary.

Furthermore, it is not enough to know if an obliquity cycle contains an interglacial or not without any difference between the onset of a new interglacial or a continued interglacials. If this difference is ignored (as suggested by the notion of the reviewer, that 23 out of 25 obliquity cycles contain an interglacial or continued interglacial), the whole approach on “finding a simple rule to determine which insolation cycles lead to interglacials” (title of T17) becomes obsolete.

With respect to the mentioned uncertainty of 20 m sea-level equivalent in the PIGS 2016 definition of interglacials we like to reply the following:

First, this would in the deconvolution of de Boer et al 2014 translate in to an uncertainty in $\delta^{18}\text{O}_{\text{sw}}$ of $\sim 0.2\text{‰}$ or of $\delta^{18}\text{O}_{\text{benthic}}$ of $\sim 0.4\text{‰}$. Interestingly, no such uncertainty in the definition of interglacials based on $\delta^{18}\text{O}$ has yet been applied in T17. It therefore seems unreasonable to us to apply such a strict and wide uncertainty here, otherwise any comparison between T17 and our studies is misleading.

Second, (and more important), in PIGS 2016 it is written that an interglacial is an interval within which the distribution of Northern Hemisphere ice resembled the present (0 ± 20 m sea level equivalent), i.e., there was little Northern Hemisphere ice outside Greenland. Thus, we understand these 20 m of sea level not as an uncertainty to this criteria, but as a suggestion how “little” the Northern Hemisphere ice outside Greenland should be. Thus, this 20 m is another threshold, now in the sea level variable, chosen to define interglacials, but to our understanding not an uncertainty in its strict sense.

Nevertheless, we thank the reviewer for this in detail comparison on the differences of our results with T17, which was to some extent so far missing in the draft. We have now included them in the revision. Furthermore, we extended the discussion, that the binary view of a glacial or interglacial situation is probably too simplistic. A new figure comparing normalized time series of LR04 with northern hemispheric ice volume outside of Greenland are compared with our binary index of glacials vs interglacials in support for this statement.

So where does this leave us with respect to the present MS? It has long been informally observed (on the basis of benthic isotope records) that pure 41kyr glacial cycles appear around the interval 1.6-1.25 Myr ago, while before that there are many instances where glacials were skipped, giving rise to continued interglacials. I would suggest that the important contribution of the present MS is that it draws attention to the fact that the early Pleistocene before 1 Ma is not one uniform interval. The discussions of insolation metrics and land ice albedo feedbacks may be interesting, but detract from the main focus of the study, which should be the small changes in NH ice volume before 1.6 Ma leading to skipped glacials.

Our reply: We agree that too many facts might have led to the distraction of the reader. We have therefore leave out a lot of the details on different insolation metrics, and of the land ice albedo feedback. However, we still think that some of its findings (e.g. which latitude is important?) are of interest and these details are kept in the draft, but in a condensed version in the discussion only without any figures.

In a pioneering modelling exercise over 20 years ago, in Berger et al. (1999, QSR) pointed out that a striking feature of the simulated ice volume with a linearly decreasing CO₂ scenario over the past 3Ma is the reversal from a state of interglacial dominance in the early Pleistocene to glacial prevailing in late Pleistocene. Clearly something changed in the climate system sometime around 1.6 Ma and I would suggest that the authors focus their discussion in that direction in a revised MS, rather than the suggestion that early Pleistocene should not be viewed as the 41-kyr world, which is not supported upon closer examination.

Chronis Tzedakis

Our reply: One has to be careful about the precise wordings: What we find is, that “the onset of new interglacials when defined by land ice distribution in the northern hemisphere outside of Greenland is not following a 41-kyr periodicity early in the Pleistocene”. This is not the same as “early Pleistocene should not be viewed as the 41-kyr world” written by the reviewer. In the underlying climate records this 41-kyr periodicity is still (and will always be) contained.

We have taken up all points suggested by the reviewer, which suggest that our classification of interglacials is uncertain. However, this still leaves us with different pattern of interglacials than T17. If we consider the unclear cases as uncertain (2–3 cases, MIS 61, MIS 71, maybe MIS 65),

we would still have 4–5 obliquity cycles in the Early Quaternary (2.6–1.6 Myr BP) with difference in the detection of the inset of new interglacials between both approaches. A way forward, in our view, might be to put not too much weight on such classification schemes, since they are due to chosen thresholds (including their uncertainties) prone to ambiguity.

We appreciate the mentioning of the Berger et al 1999 paper, that is certainly one of the earlier modelling approaches of ice volume changes across the Quaternary, although with a simpler model than de Boer et al. (2014) or Willeit et al. (2019). It is nevertheless now included in the list of previous efforts on this matter in the introduction.

All in all, we have to thank Chronis Tzedakis for his very detailed review, that certainly helped to sharpen the message of our draft.

*References

- de Boer, B., Lourens, L. J., and van de Wal, R. S.: Persistent 400,000-year variability of Antarctic ice volume and the carbon cycle is revealed throughout the Plio-Pleistocene, *Nature Communications*, 5, 2999, doi:10.1038/ncomms3999, 2014.
- Dyez, K. A., Hönisch, B., and Schmidt, G. A.: Early Pleistocene Obliquity-Scale pCO₂ Variability at ~1.5 Million Years Ago, *Paleoceanography and Paleoclimatology*, 33, 1270–1291, doi:10.1029/2018PA003349, 2018.
- Elderfield, H., Ferretti, P., Greaves, M., Crowhurst, S., McCave, I. N., Hodell, D., and Piotrowski, A. M.: Evolution of Ocean Temperature and Ice Volume Through the Mid-Pleistocene Climate Transition, *Science*, 337, 704–709, doi:10.1126/science.1221294, 2012.
- Ford, H. L. and Raymo, M. E.: Regional and global signals in seawater δ¹⁸O records across the mid-Pleistocene transition, *Geology*, 48, 113–117, doi:10.1130/G46546.1, 2019.
- Huybers, P.: Glacial variability over the last two million years: an extended depth-derived age model, continuous obliquity pacing, and the Pleistocene progression, *Quaternary Science Reviews*, 26, 37–55, doi:10.1016/j.quascirev.2006.07.013, 2007.
- Past Interglacials Working Group of PAGES: Interglacials of the last 800,000 years, *Reviews of Geophysics*, 54, 162–219, doi:10.1002/2015RG000482, 2016.
- Rohling, E. J., Foster, G. L., Grant, K. M., Marino, G., Roberts, A. P., Tamisiea, M. E., and Williams, F.: Sea-level and deep-sea-temperature variability over the past 5.3 million years, *Nature*, 508, 477–482, doi:10.1038/nature13230, 2014.
- Sosdian, S. and Rosenthal, Y.: Deep-Sea Temperature and Ice Volume Changes Across the Pliocene-Pleistocene Climate Transitions, *Science*, 325, 306–310, doi:10.1126/science.1169938, 2009.
- Tzedakis, P. C., Crucifix, M., Mitsui, T., and Wolff, E. W.: A simple rule to determine which insolation cycles lead to interglacials, *Nature*, 542, 427–432, doi:10.1038/nature21364, 2017.
- Willeit, M., Ganopolski, A., Calov, R., and Brovkin, V.: Mid-Pleistocene transition in glacial cycles explained by declining CO₂ and regolith removal, *Science Advances*, 5, eaav7337, doi:10.1126/sciadv.aav7337, 2019.

REVIEWER COMMENTS

Reviewer #1 (Remarks to the Author):

The authors have now in their third round done a good job in accommodating to the reviews. I am still somewhat skeptic, but I think the paper now will contribute constructively to the ongoing scientific discussion of categorizing the glacial cycles.

Best regards,
Peter Ditlevsen

Reviewer #2 (Remarks to the Author):

Reviewing this manuscript again, I am pleased to see the change which has occurred since the beginning of the process and I think the new focus has improved on the key messages which are delivered by the paper. However, given the substantial rewrite, I now find that some parts of the discussion are once again difficult to comprehend and will require some clarification for the readership of Nature Communications prior to publication.

I have two major points to address and several minor comments listed below:

1.

Line 93 and 222-224: Given that so much of the result depends upon this point, I think it warrants more discussion, that has jumped out at me in this revised manuscript. How might inaccuracies in the stack affect your results? Are the main conclusions still robust? Do the authors believe that the stack is accurate? Would any of the results change if the more up-to-date probSTACK of Ahn et al. 2017 were used instead? Dynamics and Statistics of the Climate System, 2017, 1–16 doi:10.1093/climsys/dzx002 Advance Access Publication Date: 26 June 2017 Research Article

2.

Lines 136-137: I'm not sure what good these means and uncertainties do here. Given that we know that these cycles are at least apparently quasi-obliquity driven. A more helpful metric to quote here might be interglacials expected: X, interglacials observed: Y, missing interglacials: X-Y. Perhaps using an 'expected' frequency of 41, 41 and 82 kyrs respectively for the 3 divisions. As currently written the reader is lead to believe a mode of 60 kyrs which may vary between 20 and 100 kyrs which I don't think is the point the authors are trying to make. Maybe pdf distributions or probability maxima would show this more effectively, i.e. can you see bimodality in 41 and 82 kyrs durations. Could these skips be directly related to problems with aliasing on LR04? I.e. are the thresholds simply not met due to age model mismatched between the constituent cores, or are there significant appearances of less extreme glacials and interglacials.

Minor points

The first sentence of the abstract is quite a jump, it requires familiarity with the body of literature which comes before and is not for the average reader. Please address this to be more contextual.

Line 20-21: I find this sentence to be a bit misleading. I would suggest something like... 'Alternatively, our definition, based upon interpreted land ice distribution and is suitable for the last 1.6 Myr, does not suit the differing ice dynamics of the early Quaternary.

Line 29: sometimes doesn't fit here, would exchange for possibly

Line 73: How is this definition objective, please expand.

Line 106: 'skipped terminations' should be in inverted commas as it is terminology created and dependent on your results.

Line 113: Why? Expand.

Line 118: I do not understand this sentence with 'temporally changing thresholds'.

Line 145: I do not understand the sentence between brackets here, please clarify.

Line 180: I think this point is crucial, and applies not only to the early Quaternary but also the Pliocene and broader Neogene where this language is used frequently.

Line 201: climate and/or CO2 could be the driver?

Line 216-221: I don't understand the reasoning behind this conclusion drawn from this one study. I'm not convinced this section is required. The study claims the importance of CO2 but confirms it is a feedback? Could it still not be the trigger, as is shown in several other modelling and data studies, where CO2 is lowered through enhanced weathering or ocean drawdown?

Line 227: would be useful to clarify here that the figure is ice volume.

Line 228: remove 'only'

Line 231: Is it a unique feature? There are potentially several such candidates for these advances. How can this be reconciled with your results? Where is the understanding gap?

Lines 248-250: Merge to be one sentence separated by a comma. I am not totally familiar with this record, but I believe that those episodes are fairly shortlived and may be due to the additional noise on this data set. I would suggest ending this section with ... 'and it puts the magnitude of sea level change implied into question'

Line 265: swap 'changes' for 'extent' for clarity.

Line 273: Missing I in 'In'.

Line 277: retrospective

Line 282: and the individual records involved in the analysis. E.g. LR04.

Line 330: It strikes me as strange that the GL ice sheet has 4 times the resolution of the other NH ice bodies, when it is the most stable of them. Could this impact on your results? It seems that the Eurasian ice sheet in particular may behave differently when restricted to a 40km by 40km resolution.

Reviewer #3 (Remarks to the Author):

K&vdW have made a great effort to focus the manuscript and address the reviewers' comments. But on the issue of the pattern of Quaternary interglacials their views have become, if anything, a little more entrenched. While a plurality of views is always a more productive way forward in science, I worry that by calling for "an irregular pattern of interglacials...across most of the last 2.6 Myr", the wrong message is being conveyed.

The authors also suggest that a binary division into glacials and interglacials is too restrictive. I agree, any nomenclature is a simplification, but that always depends on its heuristic value. In terms of climatostratigraphical and chronostratigraphical nomenclature, the binary division of the Quaternary into glacials and interglacials has served well the purpose of dividing physical sediment sequences into geological climate units and also of recording the passage of time in rocks, respectively.

However, it is somewhat misleading to suggest that this has informed "a binary view of climate", when more nuanced approaches have been in use, depending on the question asked. Within the stratigraphic nomenclature, glacials contain stadials and interstadials. A more granular approach has also been applied to interglacials. Thus, T17 distinguished between interglacials, continued interglacials and interstadials. Since the motivation was to assess when complete deglaciations occurred, "continued interglacials" were neutral because in the absence of a major glacial advance, there was no deglaciation to speak of. But continued interglacials are still interglacials...

In fact, K&vdW themselves introduce two types of interglacials: "new interglacials" and, by extension, "not new interglacials" (which are the same as the continued interglacials). Nothing wrong with that formulation, unless it starts conveying the impression that obliquity peaks with continued interglacials were skipped, as for example: "...the large number of obliquity cycles without new interglacials..." [l. 148], or "...the so-called

41-kyr world prior to the MPT would not be a period with regularly appearing interglacials..." [l. 307-308].

I suspect the authors would reply that by using the qualifiers "new interglacials" and "appearing interglacials" their statements are strictly-speaking correct, but this is not about a semantic argument, but about conveying a clear message. If the impression is given that continued (or not new) interglacials represent skipped obliquity cycles in the manner of the 100-kyr world, then this is incorrect; the four continued interglacials represent episodes where there was little NH ice outside Greenland, so the obliquity cycle during each of these periods was associated with an interglacial.

That leaves four skipped interglacials (MIS 61, 65, 71 and 77), and I am still not convinced that all of them were skipped (but we can agree to disagree on some of them, until more data are available). However, arguing for "an irregular pattern of interglacials...across most of the last 2.6 Myr" and specifically before the MPT on the basis of four skipped obliquity peaks out of 25 seems to me an exaggerated claim. Moreover, conflating the pattern before 1.6 Myr BP to that after 1 Myr BP, where 12 out of 25 obliquity peaks did not have an interglacial, is just not an accurate representation of the situation.

So how can we move forward? I think the solution lies in Supplementary Fig. 1c. If the same thresholds are applied to both the late and early parts of the Pleistocene, then there are many intervals before 1.6 Myr BP (and especially before 2 Myr BP) with extended persistence of interglacial conditions and only modest increases in ice volume, according to the model. This would argue for a different state of the climate system before 2 Myr BP, which in turn might require either a different set of thresholds and definitions or a rethink of the nature of glacial-interglacial cycles in that period. I am sorry to keep repeating this, but the answer is staring us in the face in Suppl. Fig. 1: what is different before 2 Myr BP is the relative absence of glacials...

So, the world before 2 Myr BP of missed glacials and the world after 1 Myr BP of missed interglacials, were very different, and conflating the two in order to say that there was an irregular pattern of interglacials is just missing the point.

I think that the paper can be recast with minor modifications to tell this story, but I leave this to the editor (and the authors) to decide the most appropriate way forward.

Specific comments

Lines 17-19. "Such findings would imply that the appearance of interglacials in the Quaternary as obliquity driven with a growing influence of land ice volume on the timing of deglaciations during the later part might be too simple." Since the preceding statement in the abstract on irregular interglacial patterns is not accurate, this doesn't follow either.

Lines 36-37: "is increased by the length of the previous glacial period" please change to "decreases with the amount of time since the previous deglaciation, which leads to the accumulation of ice-sheet instability".

Line 49 starts with "Alternatively", but there is no counterpoint preceding it. This could be rectified in the previous paragraph by drawing attention to the fact that instead of invoking a change in ice-sheet dynamics that led to larger ice-sheets, which in turn led to more cycles being skipped, T17 argued that a rise in the deglaciation threshold over the MPT led to more obliquity cycles being skipped, which in turn led to large ice sheets.

Fig. 1. The yellow band ~1.16 Myr BP should be removed, as it suggests that the second part of MIS 35 was not associated with an interglacial. This is not correct, as the obliquity cycle at that time is associated with a continued interglacial. Again, this is part of a more general tendency in the MS to view continued interglacials as somehow representing skipped obliquity cycles.

Chronis Tzedakis

Response to Reviews

Reviewer #1:

The authors have now in their third round done a good job in accommodating to the reviews. I am still somewhat skeptic, but I think the paper now will contribute constructively to the ongoing scientific discussion of categorizing the glacial cycles.

Best regards, Peter Ditlevsen

Our reply: We thank Peter Ditlevsen for reviewing and for agreeing with the publication of our draft.

Reviewer #2:

Reviewing this manuscript again, I am pleased to see the change which has occurred since the beginning of the process and I think the new focus has improved on the key messages which are delivered by the paper. However, given the substantial rewrite, I now find that some parts of the discussion are once again difficult to comprehend and will require some clarification for the readership of Nature Communications prior to publication.

Our reply: We also thank reviewer #2 for his/her efforts in reviewing our draft and the constructive remarks given here.

I have two major points to address and several minor comments listed below:

2.1. Line 93 and 222-224: Given that so much of the result depends upon this point, I think it warrants more discussion, that has jumped out at me in this revised manuscript. How might inaccuracies in the stack affect your results? Are the main conclusions still robust? Do the authors believe that the stack is accurate? Would any of the results change if the more up-to-date probSTACK of Ahn et al. 2017 were used instead? Dynamics and Statistics of the Climate System, 2017, 116 doi:10.1093/climsys/dzx002 Advance Access Publication Date: 26 June 2017 Research Article

Our reply: The mean of the probSTACK and the original LR04 benthic d18O stack for the last 2.6 Myr are very similar. Only around 1.8–1.9 Myr BP (MIS 65–71) both stacks differ a bit more, potentially affecting the detection of interglacials here. Apart from that we do not see that our results would change, if repeated with the new stack. The fact that differences between LR04 and probSTACK in the Quaternary are so small — both have been published 12 years apart — is an indication that the stacks are already very robust. However, significant improvements in the future, which might lead to some major changes can never be excluded. We add these details on probSTACK in the discussion of uncertainties.

2.2. Lines 136-137: I'm not sure what good these means and uncertainties do here. Given that we know that these cycles are at least apparently quasi-obliquity driven. A more helpful metric to quote here might be interglacials expected: X, interglacials observed: Y, missing interglacials: X-Y. Perhaps using an "expected" frequency of 41, 41 and 82 kyrs respectively for the 3 divisions. As currently written the reader is lead to believe a mode of 60 kyrs which may vary between 20 and 100 kyrs which I don't think is the point the authors are trying to make. Maybe pdf distributions or probability maxima would show this more effectively, i.e. can you see bimodality in 41 and 82 kyrs durations. Could these skips be directly related to problems with aliasing on LR04? I.e. are the thresholds simply not met due to age model mismatched between the constituent cores, or are there significant appearances of less extreme glacials and interglacials.

Our reply: Our idea to calculate these means was an analogy to the 100-kyr periodicity which is found after the MPT, although the underlying dynamics are multitudes of 41 kyr (thus either 2 or 3 obliquity cycles). We see, that as written so far this point does not come across. We therefore changed the text here in two ways: first, we expand on this analogy with the potentially misleading 100-kyr periodicity, and second, also add what has been suggested here: naming how much from the obliquity cycles (expected interglacials) have (realized) interglacials. The text now read as "The calculated mean ($\pm 1\sigma$) interglacial return times would be 60 (± 22) kyr, 47 (± 19) kyr, and 79 (± 24) kyr for the pre-MPT, MPT, and post-MPT, respectively. However, similarly as the average 100-kyr periodicity in the post-MPT, that according to the idea of an obliquity driven system with skipped termination causing either a ~ 80 or ~ 120 kyr return time of interglacials, these mean values here also suggest some misleading average periodicities. A more helpful way seems to be to count the realized new interglacials with respect to the potential new interglacials, with the later being identical to the obliquity cycles. Following this idea we find an new interglacial realization rate of 67% in the

pre-MPT (16 out of 24, MIS 55–101), of 88% in the MPT (14 out of 16, MIS 27–53), and of 52% in the post-MPT (12 out of 23, MIS 1–25).”

Minor points:

- 2.3. The first sentence of the abstract is quite a jump, it requires familiarity with the body of literature which comes before and is not for the average reader. Please address this to be more contextual.
Our reply: The abstract starts now as: “Glacial/interglacial dynamics during the Quaternary have been suggested to be mainly driven by obliquity with its 41-kyr periodicity including some irregularities during the last 1 Myr that resulted in on average 100-kyr cycles.”
- 2.4. Line 20-21: I find this sentence to be a bit misleading. I would suggest something like... “Alternatively, our definition, based upon interpreted land ice distribution and is suitable for the last 1.6 Myr, does not suit the differing ice dynamics of the early Quaternary.”
Our reply: Following the previous comment we reframed the whole abstract in order to make it more useful for a reader not familiar with the topic. This together with the suggested word count of 150 words we therefore had to shorten on this sentence to “Alternatively, our land ice-based definition of interglacials needs revision if applied to the entire Quaternary.”
- 2.5. Line 29: sometimes doesn’t fit here, would exchange for possibly
Our reply: Done.
- 2.6. Line 73: How is this definition objective, please expand.
Our reply: Describing this definition “objective” has been taken from T17. By reevaluating this sentence we believe this should be reframed differently. We now write “When applied for the last 800 kyr this definition has been shown to be robust against a range of definitions and thresholds”, citing PIGS-of-PAGES 2016.
- 2.7. Line 106: ‘skipped terminations’ should be in inverted commas as it is terminology created and dependent on your results.
Our reply: Done.
- 2.8. Line 113: Why? Expand.
Our reply: We changes this sentence to “In order to minimize the number of obliquity cycles without new interglacials we had to lower the thresholds by which we detect interglacials for earlier time periods in the Quaternary.” This rational has already been mentioned in the methods section.
- 2.9. Line 118: I do not understand this sentence with temporally changing thresholds.
Our reply: Changed to “The obliquity cycle without a new interglacial around MIS 61 is the only one that is situated in the time window in which the thresholds are changing.”
- 2.10. Line 145: I do not understand the sentence between brackets here, please clarify.
Our reply: This sentence in brackets is changed to: “Here, effective energy is peak caloric summer insolation plus a term related to the elapsed time since the last full deglaciation that accounts for the increasing ice sheet instability.”
- 2.11. Line 180: I think this point is crucial, and applies not only to the early Quaternary but also the Pliocene and broader Neogene where this language is used frequently.
Our reply: We extended this sentence by “and potentially also to earlier periods during the Neogene.”
- 2.12. Line 201: climate and/or CO2 could be the driver?
Our reply: Changed.
- 2.13. Line 216-221: I don’t understand the reasoning behind this conclusion drawn from this one study. I’m not convinced this section is required. The study claims the importance of CO2 but confirms it is a feedback? Could it still not be the trigger, as is shown in several other modelling and data studies, where CO2 is lowered through enhanced weathering or ocean drawdown?
Our reply: We understand that this paragraph was slightly confusing. We therefore reframed

the final sentence into “This illustrates, that it is not yet clear whether CO₂ acted as trigger or feedback during the glacial/interglacial evolution across the MPT.”

- 2.14. Line 227: would be useful to clarify here that the figure is ice volume.
Our reply: Done.
- 2.15. Line 228: remove only
Our reply: Done.
- 2.16. Line 231: Is it a unique feature? There are potentially several such candidates for these advances. How can this be reconciled with your results? Where is the understanding gap?
Our reply: We deleted the word “unique”. Right now, such early glacial advances as far south as 38°N can not be reconciled with our present knowledge on climate change. Either we are missing some details on radiative forcing that has been active at that time, or these data need to be reinterpreted, or ice sheets models used for simulating these events have been too simplistic. We include these details in the draft.
- 2.17. Lines 248-250: Merge to be one sentence separated by a comma. I am not totally familiar with this record, but I believe that those episodes are fairly shortlived and may be due to the additional noise on this data set. I would suggest ending this section with “... and it puts the magnitude of sea level change implied into question”.
Our reply: Sentences are merged, and the ending is changed as suggested.
- 2.18. Line 265: swap “changes” for “extent” for clarity.
Our reply: Done.
- 2.19. Line 273: Missing I in In.
Our reply: Done.
- 2.20. Line 277: retrospective
Our reply: Done.
- 2.21. Line 282: and the individual records involved in the analysis. E.g. LR04.
Our reply: This detail has now been included.
- 2.22. Line 330: It strikes me as strange that the GL ice sheet has 4 times the resolution of the other NH ice bodies, when it is the most stable of them. Could this impact on your results? It seems that the Eurasian ice sheet in particular may behave differently when restricted to a 40km by 40km resolution.
Our reply: Simulating the Greenland ice sheet requires a higher resolution to resolve the narrow ablation zone which is even in the current climate not more than 40 km on average, so too coarsely resolved simulations lead to simulations without ablation. This does not happen for the North American and Eurasian ice bodies with large Southern margins.

Reviewer #3:

K&vdW have made a great effort to focus the manuscript and address the reviewers' comments. But on the issue of the pattern of Quaternary interglacials their views have become, if anything, a little more entrenched. While a plurality of views is always a more productive way forward in science, I worry that by calling for "an irregular pattern of interglacials...across most of the last 2.6 Myr", the wrong message is being conveyed.

Our reply: We changed the title into "Interglacials of the Quaternary defined by northern hemispheric land ice distribution outside of Greenland". In the abstract "irregular pattern of interglacials" is replaced by the fraction of obliquity cycles with realized new interglacials for the early, mid and late Quaternary.

The authors also suggest that a binary division into glacials and interglacials is too restrictive. I agree, any nomenclature is a simplification, but that always depends on its heuristic value. In terms of climatostratigraphical and chronostratigraphical nomenclature, the binary division of the Quaternary into glacials and interglacials has served well the purpose of dividing physical sediment sequences into geological climate units and also of recording the passage of time in rocks, respectively.

Our reply: The problem of thresholds and definitions has already been mentioned in the PIGS of PAGES (2016) review on interglacials.

However, it is somewhat misleading to suggest that this has informed "a binary view of climate", when more nuanced approaches have been in use, depending on the question asked. Within the stratigraphic nomenclature, glacials contain stadials and interstadials. A more granular approach has also been applied to interglacials. Thus, T17 distinguished between interglacials, continued interglacials and interstadials. Since the motivation was to assess when complete deglaciations occurred, continued interglacials were neutral because in the absence of a major glacial advance, there was no deglaciation to speak of. But continued interglacials are still interglacials

Our reply: Following the review, we tone down on this issue: In the abstract this sentence on the binary view of climate is deleted and more room is given in explaining the problem - the so-called Mid-Pleistocene Transition.

In fact, KvdW themselves introduce two types of interglacials: "new interglacials" and, by extension, "not new interglacials" (which are the same as the continued interglacials). Nothing wrong with that formulation, unless it starts conveying the impression that obliquity peaks with continued interglacials were skipped, as for example: "...the large number of obliquity cycles without new interglacials..." [l. 148], or "...the so-called 41-kyr world prior to the MPT would not be a period with regularly appearing interglacials..." [l. 307-308].

Our reply: A new interglacial starts with a termination. We therefore think, this difference "new interglacials" vs "not new interglacials" is meaningful. It however follows out of T17, in which it is shown how insolation leads to new interglacials (thus to a termination).

I suspect the authors would reply that by using the qualifiers "new interglacials" and "appearing interglacials" their statements are strictly-speaking correct, but this is not about a semantic argument, but about conveying a clear message. If the impression is given that continued (or not new) interglacials represent skipped obliquity cycles in the manner of the 100-kyr world, then this is incorrect; the four continued interglacials represent episodes where there was little NH ice outside Greenland, so the obliquity cycle during each of these periods was associated with an interglacial.

Our reply: Continued or not new interglacials are one of two ways by which an obliquity cycles

contains no termination, the other way is a skipped termination. Both are clearly distinguished by the colors of the labels of MIS (red vs blue) in Fig 3. We never use the term “skipped obliquity”. We never argued that the four continued interglacials represent episodes where there was little NH ice outside Greenland (MIS 55, 79, 93, 101, red labels in Fig 3) are no interglacial as suggested. The point is about do we have a new interglacial = a termination.

That leaves four skipped interglacials (MIS 61, 65, 71 and 77), and I am still not convinced that all of them were skipped (but we can agree to disagree on some of them, until more data are available). However, arguing for “an irregular pattern of interglacials...across most of the last 2.6 Myr” and specifically before the MPT on the basis of four skipped obliquity peaks out of 25 seems to me an exaggerated claim. Moreover, conflating the pattern before 1.6 Myr BP to that after 1 Myr BP, where 12 out of 25 obliquity peaks did not have an interglacial, is just not an accurate representation of the situation.

Our reply: Following a comment of reviewer #2 we have now calculated the new interglacial realization rate, that is 67% in the pre-MPT (16 out 24 for MIS 55-101), of 88% in the MPT (14 out of 16 for MIS 27-53), and of 52% in the post-MPT (12 out 23 for MIS 1-25). This is a more qualitative measure of irregularity, which we will now name more prominently in the abstract and the conclusions.

So how can we move forward? I think the solution lies in Supplementary Fig. 1c. If the same thresholds are applied to both the late and early parts of the Pleistocene, then there are many intervals before 1.6 Myr BP (and especially before 2 Myr BP) with extended persistence of interglacial conditions and only modest increases in ice volume, according to the model. This would argue for a different state of the climate system before 2 Myr BP, which in turn might require either a different set of thresholds and definitions or a rethink of the nature of glacial-interglacial cycles in that period. I am sorry to keep repeating this, but the answer is staring us in the face in Suppl. Fig. 1: what is different before 2 Myr BP is the relative absence of glacials...

Our reply: We have taken up on this suggestions, implying that Suppl Fig 1 is more important than we have thought. We have therefore moved it to the main text and extended on the discussion if one definition of interglacials can be used across the Quaternary at the end of the results: “Actually, our definition of interglacials clearly illustrates a shift in the climate system, from an interglacial dominated period in the early to an glacial dominated period in the late Quaternary. This becomes especially clear when we base our findings on constant thresholds (e.g. Fig. 2c).” and in the conclusions: “Following our definition the appearance of interglacials clearly shifts across the Quaternary, from an interglacial-dominated to glacial-dominated climate.”

So, the world before 2 Myr BP of missed glacials and the world after 1 Myr BP of missed interglacials, were very different, and conflating the two in order to say that there was an irregular pattern of interglacials is just missing the point.

Our reply: In our view there are still two ways how one can understand our results: (1) We apply our definition of interglacials and find irregular patterns early and late in the Quaternary. (2) A unified definition of interglacials across the Quaternary is not possible. Both views are clearly mentioned in the draft, already in the abstract and in the conclusions. We do not agree with the comment above, which suggests that view (1) is entirely wrong. We think both views are potentially possible. In that way, I believe we can only summarize this conflict, as done by the reviewer already, that we agree to disagree.

I think that the paper can be recast with minor modifications to tell this story, but I leave this to the editor (and the authors) to decide the most appropriate way forward.

Our reply: We have included most of the suggestions, which indeed helped to strengthen the paper.

Specific comments:

- 3.1. Lines 17-19. "Such findings would imply that the appearance of interglacials in the Quaternary as obliquity driven with a growing influence of land ice volume on the timing of deglaciations during the later part might be too simple." Since the preceding statement in the abstract on irregular interglacial patterns is not accurate, this doesn't follow either.

Our reply: The preceding statement has been replaced now. The mentioned statement is a direct consequence of where we disagree with the reviewer (see above), in that we think one might still follow the definition of interglacials suggested here and would then come to these conclusions. To reduce our study to the outcome, that one generic definition of interglacials is not applicable across the whole Quaternary (which would imply that the sentence above is not accurate) is in our view only an alternative way of the possible interpretations.

- 3.2. Lines 36-37: "is increased by the length of the previous glacial period" please change to "decreases with the amount of time since the previous deglaciation, which leads to the accumulation of ice-sheet instability".

Our reply: Done.

- 3.3. Line 49 starts with "Alternatively", but there is no counterpoint preceding it. This could be rectified in the previous paragraph by drawing attention to the fact that instead of invoking a change in ice-sheet dynamics that led to larger ice-sheets, which in turn led to more cycles being skipped, T17 argued that a rise in the deglaciation threshold over the MPT led to more obliquity cycles being skipped, which in turn led to large ice sheets.

Our reply: Instead of putting more details to the previous paragraph, as suggested here, we have chosen to shorten the sentence, which started with "Alternatively". The first part ("Alternatively to those orbital theories") is now deleted to avoid confusion.

- 3.4. Fig. 1. The yellow band ~ 1.16 Myr BP should be removed, as it suggests that the second part of MIS 35 was not associated with an interglacial. This is not correct, as the obliquity cycle at that time is associated with a continued interglacial. Again, this is part of a more general tendency in the MS to view continued interglacials as somehow representing skipped obliquity cycles.

Our reply: We disagree. In T17 the obliquity cycle around ~ 1.16 Myr BP (yellow band here) does not contain the onset of a new interglacial, which is the definition of where we put yellow bands in this figure (although we clarified the caption in this respect). Furthermore, if we would have to remove the yellow band here, it would also need to be removed in figure 2. Our interpretation of T17 is focussing on the main aspect: does the obliquity cycle contain a new interglacial or not. Our understanding is, that if this is not the case we either have a skipped termination or a continued interglacial, both are clearly distinguished in Fig 3 (color of labelled MIS either red (continued interglacial, including this band here identified as MIS34), or blue (skipped termination). We therefore can not follow the comment, that we have the tendency to view continued interglacials as somehow representing skipped terminations. However, we have to acknowledge, that we have been not entirely correct when describing the yellow bands in the text (page 5). This has now been changed to "For times after 1.6 Myr BP, our approach detects nearly the same obliquity cycles as in T17 with the onset of new interglacials. This includes 13 obliquity cycles without detected interglacials, of which 12 are known as 'skipped terminations', while one (including MIS 34) is a continued interglacial." We are aware, that what the reviewer called "second part of MIS 35" was in Fig 3 labelled "MIS 34". That was done on purpose to not confuse it with MIS 35 around 1.2 Myr BP, in which a new interglacial has started in the previous obliquity cycle.

Our reply: We thank Chronis Tzedakis for his insights on T17 and his review here.

REVIEWER COMMENTS

Reviewer #2 (Remarks to the Author):

I thank the authors for persisting through this many rounds of revision, and congratulate them on a much improved manuscript.

I am happy to accept publication in this current form.

Best wishes,
Thomas Chalk

Reviewer #3 (Remarks to the Author):

It is certainly not the role of a reviewer to impose one's own views on the authors of a MS. One can agree to disagree, provided that the approach followed is clearly stated. K&vdW have produced a substantially improved version that goes a long way in addressing the issues raised. Some points remain that require some minor modification, as I will explain below, but the MS is now near publication.

Reading the new version and the authors' response to comments, I am finally able to see where views diverge. As the authors state, their analysis is about 'terminations'. The simple rule of T17 also considered the same thing, 'complete deglaciations' leading to interglacials. However, one difference is that T17 considered continued interglacials as neutral, whereas their status in K&vdW is more ambiguous. In their rebuttal K&vdW say:

"Continued or not new interglacials are one of two ways by which an obliquity cycle contains no termination, the other way is a skipped termination. Both are clearly distinguished by the colors of the labels of MIS (red vs blue) in Fig 3. We never use the term "skipped obliquity". We never argued that the four continued interglacials represent episodes where there was little NH ice outside Greenland (MIS 55, 79, 93, 101, red labels in Fig 3) are no interglacial as suggested. The point is about do we have a new interglacial = a termination."

This may be correct, but 'continued interglacials' are not that clearly denoted in the figures. It is true that they appear in the red MIS numbers in Fig. 4 (not in Fig. 3), but this is difficult to pick out in a very busy figure, and in the rest of the figures they are denoted by yellow bands, the same as skipped terminations. But the two are not the same. More importantly, their "new interglacial realization rate" again excludes continued interglacials, so that in the pre-MPT we have 16 new interglacials out of 24 obliquity cycles (67%), but if we include the continued interglacials we have 20 interglacials out of 24 obliquity cycles (83%).

Again, strictly-speaking, the authors are correct, because they talk about new interglacials. But I worry (in fact, I am almost sure) that this would be misinterpreted by readers who are not so well versed in the finer points of nomenclature. I have a few modest suggestions:

1. in the text, use 'terminations' more often instead of 'interglacials' or 'new interglacials'. This is already the case in the abstract (line 19), but this could also be applied to lines 21, 187, 332, 334.
2. At the end of the sentence regarding the new interglacial realization rate (lines 143-146), please include a sentence making clear that continued interglacials have been treated the same as skipped terminations in this analysis, but if they were included then the number of interglacials corresponding to obliquity cycles would be higher.
3. Denote continued interglacials more clearly in Figs 2 and 3.

Specific comments:

Abstract: I am not sure if there is enough space, but the abstract should convey how this work is able to establish the amount of NH land ice outside Greenland.

Abstract: "*Our findings suggest that the proposed idea of interglacials in the Quaternary as obliquity driven with a growing influence of land ice volume on the timing of deglaciations during the last 1Myr might be too simple.*" As I have a personal interest in this, I am forced to ask: which part of this idea do the authors find too simple? All of it?

Lines 36-38: "*In this concept the amount of energy necessary to trigger a deglaciation which leads into an interglacial, decreases with the amount of time since the previous deglaciation, enhancing ice-sheet instability.*" This sentence is still not quite right. Please replace "enhancing ice-sheet instability" with "as ice-sheet instability increases".

Lines 191-193: Please refer to the work of Berger et al., (1999) who were the first to suggest the shift from interglacial-dominated to glacial-dominated climate over the course of the Quaternary (and also some undertake some minor corrections ['Quaternary' was missing from 'early']): "Actually, our definition of interglacials clearly illustrates a shift in the climate system, from an interglacial-dominated period in the early Quaternary to a glacial-dominated period in the late Quaternary, as originally proposed in ref. 28."

I would like to thank the authors for their perseverance and efforts to improve the MS.

Chronis Tzedakis

Response to Reviews

Reviewer #2:

I thank the authors for persisting through this many rounds of revision, and congratulate them on a much improved manuscript.

I am happy to accept publication in this current form.

Best wishes, Thomas Chalk

Our reply: Thanks again to Thomas Chalk for his efforts in reviewing and therefore improving our draft. We are also happy that he is happy with the improved version of the draft.

Reviewer #3:

It is certainly not the role of a reviewer to impose one's own views on the authors of a MS. One can agree to disagree, provided that the approach followed is clearly stated. K&vdW have produced a substantially improved version that goes a long way in addressing the issues raised. Some points remain that require some minor modification, as I will explain below, but the MS is now near publication.

Reading the new version and the authors' response to comments, I am finally able to see where our views diverge. As the authors state, their analysis is about "terminations". The simple rule of T17 also considered the same thing, "complete deglaciations" leading to interglacials. However, one difference is that T17 considered continued interglacials as neutral, whereas their status in K&vdW is more ambiguous. In their rebuttal K&vdW say:

"Continued or not new interglacials are one of two ways by which an obliquity cycle contains no termination, the other way is a skipped termination. Both are clearly distinguished by the colors of the labels of MIS (red vs blue) in Fig 3. We never use the term "skipped obliquity". We never argued that the four continued interglacials represent episodes where there was little NH ice outside Greenland (MIS 55, 79, 93, 101, red labels in Fig 3) are no interglacial as suggested. The point is about do we have a new interglacial = a termination."

This may be correct, but 'continued interglacials' are not that clearly denoted in the figures. It is true that they appear in the red MIS numbers in Fig. 4 (not in Fig. 3), but this is difficult to pick out in a very busy figure, and in the rest of the figures they are denoted by yellow bands, the same as skipped terminations. But the two are not the same. More importantly, their new interglacial realization rate again excludes continued interglacials, so that in the pre-MPT we have 16 new interglacials out of 24 obliquity cycles (67%), but if we include the continued interglacials we have 20 interglacials out of 24 obliquity cycles (83%).

Again, strictly-speaking, the authors are correct, because they talk about new interglacials. But I worry (in fact, I am almost sure) that this would be misinterpreted by readers who are not so well versed in the finer points of nomenclature. I have a few modest suggestions:

Our reply: We are happy to follow the suggested final improvements. In detail, we have in a few cases reasons to implement them a bit differently than suggested. See our rebuttals below.

1. in the text, use 'terminations' more often instead of 'interglacials' or 'new interglacials'. This is already the case in the abstract (line 19), but this could also be applied to lines 21, 187, 332, 334.

Our reply: Done.

2. At the end of the sentence regarding the new interglacial realization rate (lines 143-146), please include a sentence making clear that continued interglacials have been treated the same as skipped terminations in this analysis, but if they were included then the number of interglacials corresponding to obliquity cycles would be higher.

Our reply: Done. The new sentence reads: "These rates should not be mixed with the fraction of obliquity cycles that include interglacials, since the realization rate of new interglacials decreases not only for obliquity cycles with a skipped termination but also for those with a continued interglacial."

3. Denote continued interglacials more clearly in Figs 2 and 3.

Our reply: The color-code of the MIS labelling in Figs 2 and 3 has now been used as already done in Fig 4 to distinguish between skipped terminations (blue) and continued interglacials (red). In doing so, and to make clear, that the colors of the MIS labels and of the bars marking additional interglacials according to this study means something different we also change the color of these bars from red to pink in Fig 2a and 3a. Furthermore, in doing so we realized that in Fig 3b the color code of MIS 34 was wrong. It has been marked grey, which would mark an obliquity cycle that contains no new interglacials according to T17, which has not been confirmed in our definition. In this alternative definition found in Fig 3b, we indeed also find around MIS 34 an obliquity cycle without a new interglacial.

Interestingly, the reason is now a skipped termination, while it has been a continued interglacial in Fig 3a, which contains our standard definition. This change in the causing process when changing the threshold in Fig 3 happens for a few MIS, e.g in Fig 3c all reasons for obliquity cycles without a new interglacials before (and including) MIS 34 are continued interglacials, and not skipped terminations which has been the reason in MIS 61, 65, 71, 77 in our standard definition. This finding is added with a paragraph in the result section reading:

"When our approach is based on the alternative thresholds (Fig. 3b,c) we also find that the reason for an obliquity cycle to miss a new interglacial might change. When based on the higher threshold (Fig. 3c) we only find continued interglacials (red labeled MIS), but no skipped terminations (blue labeled MIS) for times early than MIS 33 around 1.15 Myr BP. Furthermore, the result for MIS 34 largely depends on the chosen threshold. The changes in ice volume between a glacial and an interglacial period in the early Quaternary are rather small and definitions based upon them will always be prone to some arbitrariness."

Specific comments:

3.1. Abstract: I am not sure if there is enough space, but the abstract should convey how this work is able to establish the amount of NH land ice outside Greenland.

Our reply: This has been realized by adding "Deconvolving benthic $\delta^{18}\text{O}$ with a framework that includes a 3D global land ice model ..." which are 14 more words, pushing the abstract word count to 176, which hopefully is acceptable for the journal. If this becomes a word-count problem, I fear we have to delete this sentence again.

3.2. Abstract: "Our findings suggest that the proposed idea of interglacials in the Quaternary as obliquity driven with a growing influence of land ice volume on the timing of deglaciations

during the last 1Myr might be too simple.” As I have a personal interest in this, I am forced to ask: which part of this idea do the authors find too simple? All of it?

Our reply: Land ice volumes in the early Quaternary is only weakly following benthic $\delta^{18}\text{O}$ and therefore in the early Quaternary land ice volume outside of Greenland is very different when based on our interpretation. If one believes that our reconstruction of land ice volume is superior to what has been done in T17 (because our approach is based on a deconvolution of $\delta^{18}\text{O}$ that splits the temperature effect from the sea level effect and that the ice distribution is then based on physical constant land ice models) then the simple rule of T17 “interglacials in the Quaternary as obliquity driven with a growing influence of land ice volume on the timing of deglaciations during the last 1Myr” does not work anymore — and is therefore too simple. It fails in the early Quaternary, but restricted to the last 1.6 Myr it seems to bring reasonable result. In the category of being too simple falls probably also the assumption made in T17 that ice sheet instability increases with time. To our understanding (and based on processes), it is probably the case that ice sheet instability increases with size, not time (see also our reply to the next comment). Saying here in the abstract that the rule is “too simple” is motivated by the title of T17 “A simple rule to determine which insolation cycles lead to interglacials”, and condenses our findings as much as possible but necessary in a very short abstract.

- 3.3. Lines 36-38: “In this concept the amount of energy necessary to trigger a deglaciation which leads into an interglacial, decreases with the amount of time since the previous deglaciation, enhancing ice-sheet instability.” This sentence is still not quite right. Please replace “enhancing ice-sheet instability” with “as ice-sheet instability increases”.

Our reply: If changes as suggested the sentence would suggest that it is universally true that ice-sheet instability increases over time. However, we do not think this is the case. To our understanding a more general mechanism would be that ice-sheet instability increases with size, not time (e.g. see Bintanja and van de Wal (2008), ref 6). We therefore revised the sentence differently as suggested now reading: “In this concept the amount of energy necessary to trigger a deglaciation which leads into an interglacial decreases with the amount of time since the previous deglaciation as it is assumed in T17 that ice-sheet instability increases with time.”

- 3.4. Lines 191-193: Please refer to the work of Berger et al., (1999) who were the first to suggest the shift from interglacial-dominated to glacial-dominated climate over the course of the Quaternary (and also some undertake some minor corrections [Quaternary was missing from early]): Actually, our definition of interglacials clearly illustrates a shift in the climate system, from an interglacial-dominated period in the early Quaternary to a glacial-dominated period in the late Quaternary, as originally proposed in ref. 28.

Our reply: Realized as “Actually, our definition of interglacials clearly illustrates a shift in the climate system, from an interglacial-dominated period in the early Quaternary to a glacial-dominated period in the late Quaternary, a pattern which has originally been suggested from results of a simplified vertically integrated sectorial ice model without thermodynamics (ref 28)”.

I would like to thank the authors for their perseverance and efforts to improve the MS.
Chronis Tzedakis

Our reply: Again, we thank Chronis Tzedakis for his insights on T17 and his further very detailed remarks given here.

We furthermore implemented some small improvements which should clarify the messages: In detail: — In the last sentence of the abstract we replaced “our” with “the” in the sentence now reading: “Alternatively, the land ice-based definition of interglacials needs revision if applied to the entire Quaternary.”

— In the results section “realization rate” is replaced with “realization fraction” (once).

REVIEWERS' COMMENTS

Reviewer #3 (Remarks to the Author):

I am grateful to the authors for undertaking the suggested changes and happy to recommend publication.

I have just a comment for the authors, since they raised an interesting point. In their rebuttal, K&vdW say:

CT: Lines 36-38: "In this concept the amount of energy necessary to trigger a deglaciation which leads into an interglacial, decreases with the amount of time since the previous deglaciation, enhancing ice-sheet instability." This sentence is still not quite right. Please replace "enhancing ice-sheet instability" with "as ice-sheet instability increases".

K&vdW reply: If changes as suggested the sentence would suggest that it is universally true that ice-sheet instability increases over time. However, we do not think this is the case. To our understanding a more general mechanism would be that ice-sheet instability increases with size, not time (e.g. see Bintanja and van de Wal (2008), ref 6). We therefore revised the sentence differently as suggested now reading: "In this concept the amount of energy necessary to trigger a deglaciation which leads into an interglacial, decreases with the amount of time since the previous deglaciation as it is assumed in T17 that ice-sheet instability increases with time."

CT: (I am not sure you need "it is assumed in T17 that" and "with time" since that sentence and the previous one refers to T17, but be that as it may). I do not disagree with the authors that instability is linked to the size of ice sheets. Tzedakis et al. (2017; p. 429) said:

"This instability can be due to any of the following negative feedbacks on ice growth: (i) mechanical instabilities of the ice–bedrock system, enhanced calving and exposure to lower-latitude insolation as ice sheets grow; (ii) a decrease in ice-sheet albedo and an increase in ablation as a result of higher rates of dust deposition as ice sheets expand; and (iii) releases of deep-ocean CO₂ as a function of extension of the Antarctic ice sheet over continental shelves."

The size of ice sheets is critical, but time may also enter into the equation indirectly. For example, as aridity increases over time and vegetation cover decreases, more dust will accumulate on the ice sheets, reducing albedo. Ultimately, time and ice sheet size are interrelated, as ice volume increases over time during a glacial. The reason T17 used 'time' and not 'ice volume' is that they wanted to derive the simplest possible rule without any palaeoclimate data.

Chronis Tzedakis

Response to Reviews

Reviewer #3:

I am grateful to the authors for undertaking the suggested changes and happy to recommend publication.

I have just a comment for the authors, since they raised an interesting point. In their rebuttal, K&vdW say:

CT: Lines 36-38: In this concept the amount of energy necessary to trigger a deglaciation which leads into an interglacial, decreases with the amount of time since the previous deglaciation, enhancing ice-sheet instability. This sentence is still not quite right. Please replace enhancing ice-sheet instability with as ice-sheet instability increases.

K&vdW reply: If changes as suggested the sentence would suggest that it is universally true that ice-sheet instability increases over time. However, we do not think this is the case. To our understanding a more general mechanism would be that ice-sheet instability increases with size, not time (e.g. see Bintanja and van de Wal (2008), ref 6). We therefore revised the sentence differently as suggested now reading: In this concept the amount of energy necessary to trigger a deglaciation which leads into an interglacial, decreases with the amount of time since the previous deglaciation as it is assumed in T17 that ice-sheet instability increases with time.

CT: (I am not sure you need it is assumed in T17 that and with time since that sentence and the previous one refers to T17, but be that as it may). I do not disagree with the authors that instability is linked to the size of ice sheets. Tzedakis et al. (2017; p. 429) said: This instability can be due to any of the following negative feedbacks on ice growth: (i) mechanical instabilities of the icebedrock system, enhanced calving and exposure to lower-latitude insolation as ice sheets grow; (ii) a decrease in ice-sheet albedo and an increase in ablation as a result of higher rates of dust deposition as ice sheets expand; and (iii) releases of deep-ocean CO₂ as a function of extension of the Antarctic ice sheet over continental shelves.

The size of ice sheets is critical, but time may also enter into the equation indirectly. For example, as aridity increases over time and vegetation cover decreases, more dust will accumulate on the ice sheets, reducing albedo. Ultimately, time and ice sheet size are interrelated, as ice volume increases over time during a glacial. The reason T17 used time and not ice volume is that they wanted to derive the simplest possible rule without any palaeoclimate data.

Chronis Tzedakis

Our reply: We thank for the clarification, why in T17 time was chosen as variable by which ice sheets might become unstable. The reviewer is not sure if all our edits are necessary, but requests no further changes. However, we believe the sentence as it is now is most clear and leave it as it is (the addition of these further details from T17 would not help any further here).